# Heterogeneous Federated Learning: A Dual Matching Dataset Distillation Approach

## Abstract

Federated Learning (FL) often struggles with error accumulation during local training, particularly on heterogeneous data, which hampers overall performance and convergence. While dataset distillation is commonly introduced to FL to enhance efficiency, our work finds that communicating distilled data instead of models can completely get rid of the error accumulation issue, albeit at the cost of exacerbating data heterogeneity across clients. To address the amplified heterogeneity due to distilled data, we propose a novel FL algorithm termed *FedDual-Match*, which performs dual matching in the way that local distribution matching captures client data distributions while global gradient matching aligns gradients on the server. This dual approach enriches feature representations and enhances convergence stability. It proves effective for FL due to a bounded difference in the testing loss between optimal models trained on the aggregation of either distilled or original data across clients. At the same time, it can converge to within a bounded constant of the optimal model loss. Experiments on controlled heterogeneous dataset MNIST/CIFAR10 and naturally heterogeneous dataset Digital-Five/Office-Home demonstrate its advantages over the state-of-the-art methods that communicate either model or distilled data, in terms of accuracy and convergence. Notably, it maintains accuracy even when data heterogeneity significantly increases, underscoring its potential for practical applications.

## 1 Introduction

Federated learning (FL) is a distributed paradigm that enables collaborative optimization across devices while preserving data privacy (Bonawitz, 2019). It has been widely adopted in areas of healthcare (Xu et al., 2021), finance (Li et al., 2020), and the Internet of Things (IoT) (Kairouz et al., 2021). Due to the communication cost constraint, federated clients typically run multiple local training epochs before communicating with the server. However, client heterogeneity can cause weight drifts, which can be further accumulated during communication, leading to performance drop and convergence instability. To mitigate these challenges, various methods have been proposed for heterogeneous federated learning, such as regularization techniques (Li et al., 2020), weighted aggregation (Wang et al., 2020a), and personalization strategies (T Dinh et al., 2020). Despite these efforts, error accumulation persists. Recently, dataset distillation techniques have been introduced to FL to enhance efficiency (Xiong et al., 2023; Pi et al., 2023), where client data is condensed into smaller yet information-dense subsets with similar training utility. In addition to improving efficiency, in this work, we find that communicating distilled data instead of models can effectively get rid of the error accumulation, as the model can be fully optimized with aggregated distilled data on the server in a way akin to the centralized training. However, an issue that accompanies this change is that the selective extraction and integration of client data features may exacerbate heterogeneity among distilled datasets compared to the original client data (Huang et al., 2023). To delve into this issue and the underlying paradigm shift in communication, one will naturally ask the following two questions for FL:

- *How can we fully exploit distilled data and mitigate the amplified heterogeneity?*
- *Why can distilled data replace model for communication?*

To answer the first question, we propose a novel FL framework, called **FedDualMatch**, that can leverage complementary strengths of gradient and distribution matching to effectively address the

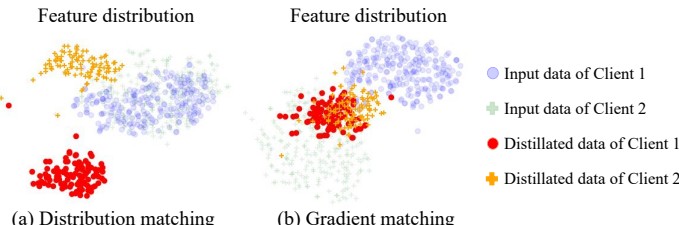

Figure 1: (a) Distribution matching amplifies the heterogeneity of distilled data between clients, while (b) gradient matching reduces the client drift.

amplified heterogeneity on distilled data. As illustrated by Fig. 1 (a), distribution matching excels at extracting data features but may exacerbate the heterogeneity issue (Shin et al., 2023). Contrastingly, gradient matching, described in Fig. 1 (b), can control the client drift (Zhao et al., 2020) and thus alleviate the distribution heterogeneity. This insight motivates us to propose *local distribution matching (LDM)* for exploiting the unique data features on the client side. It distills data by matching the mean distribution of the distilled data with that of the given data on each client in multiple layer-wise feature spaces. To improve the robustness of distillation, we also design Gaussian ball sampling (Xiong et al., 2023) with adaptive generalization error radius. However, such local data distillation amplifies the heterogeneity of distilled data between clients, exacerbating the client drift. To rectify this issue, we propose *global gradient matching (GGM)* on the server side, which constrains the global model's gradients for consistency between the aggregated distilled dataset and individual clients' distilled datasets and thus improves the convergence stability of the global model. Fig. 2 describes how the dual matching strategies are integrated into our FedDualMatch to sufficiently distill data features and meanwhile enhance convergence stability.

For the second question, we conduct a theoretical analysis of the communication of distilled datasets in terms of effectiveness and convergence. Specifically, we show that under the assumptions of *bounded distributional distance* and *bounded gradient distance*, the testing loss difference between the optimal model trained on aggregated distilled datasets and that trained on all the given datasets in the centralized setting can be bounded by a small positive constant. It means that training on the aggregated distilled dataset closely approximates the ideal centralized training, thus supporting the effectiveness of distilled data communication. At the same time, the global model can converge to the sub-optimality gap of the final model is bounded by a constant primarily determined by the bounded distributional distance.

We provide extensive experiments to evaluate the efficacy and effectiveness of FedDualMatch under controlled or natural heterogeneity. In controlled experiments with artificial heterogeneity, we set up the data distributions of MNIST and CIFAR10 via a Dirichlet distribution. Our experimental studies on them show that it outperforms those existing federated learning methods, based on either distilled data communication or traditional model communication, in terms of convergence speed and model accuracy. Further, experimental results on naturally heterogeneous datasets, such as Digit-Five and Office-Home, demonstrate that it exhibits strong stability and adaptability when handling real-world data heterogeneity. To summarize, we make the following contributions:

- We propose FedDualMatch, a novel federated learning framework that combines gradient and distribution matching for distilled dataset communication. It excels in getting rid of error accumulation while reducing client drift caused by the amplified heterogeneity from distilled data.

- We provide theoretical analysis on the effectiveness and stable convergence of distilled dataset communication.

- We conduct extensive experiments which show that our proposed algorithm outperforms the state-of-the-art on both controlled and naturally heterogeneous datasets and particularly maintains accuracy on data with increasing heterogeneity.

## 2 RELATED WORK

**Heterogeneous federated learning (HFL)**: HFL focuses on the significant challenge of non-IID data distributions across clients as a pervasive issue in real-world federated learning scenarios. To

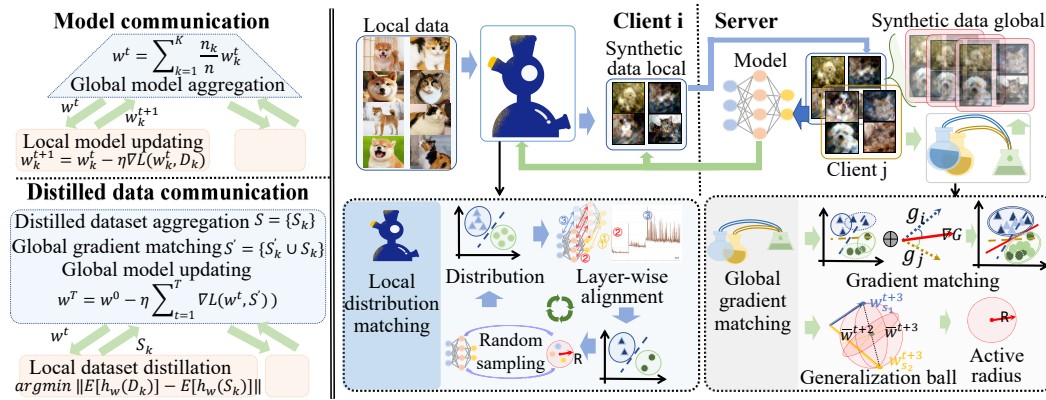

(a) Communication comparison     (b) Framework of FedDualMatch

Figure 2: (a) The replacement from model to distilled data and (b) the framework of FedDualMatch. Clients receive the global model from the server, use local distribution-matching-based dataset distillers to generate synthetic data, and send this data to the server. The server aggregates the distilled data from all clients, corrects synthetic data heterogeneity using a global gradient-matching-based dataset distiller, and updates the global model with the corrected aggregated data. The updated model is then sent back to the clients, repeating the process for federated training.

tackle this, various strategies have emerged. Personalization methods, including pFedMe (T Dinh et al., 2020), FedBN (Li et al., 2021c), and Ditto (Li et al., 2021b), adapt global models to client-specific data, in order to enhance local performance while maintaining overall generalization. Regularization approaches such as FedProx (Li et al., 2020), Scaffold (Karimireddy et al., 2020), and MOON (Li et al., 2021a) control client drift and stabilize training under heterogeneity by introducing constraints on local updates. Advanced aggregation techniques such as FedMA (Wang et al., 2020a), FedNova (Wang et al., 2020b), and FedDyn (Acar et al., 2021) improve model alignment and convergence under data heterogeneity. Additionally, model and data distillation strategies, like FedDF (Lin et al., 2020), FedMD (Li & Wang, 2019), and FedBE (Chen & Chao, 2020), enhance global model robustness by aggregating knowledge from diverse clients. Meta-learning approaches, e.g., MetaFed (Jiang et al., 2019) and Per-FedAvg (Fallah et al., 2020), leverage dynamic adaptation to varying client distributions. These methods, all doing model communication, albeit effective, just alleviate the issue of error accumulation instead of eliminating it at the source.

**Dataset-distillation-based federated learning**: Recent advancements in FL have leveraged dataset distillation via either gradient/trajectory matching or distribution matching to enhance communication efficiency and address client heterogeneity. Gradient and trajectory-based approaches, such as FedMK (Liu et al., 2022), FedSynth (Hu et al., 2022), DYNAFED (Pi et al., 2023), FedLAP-DP (Wang et al., 2023), and FEDLGD (Huang et al., 2023), focus on condensing gradient information or optimizing trajectories to streamline communication and improve convergence. In contrast, distribution-based methods, e.g., FedDM (Xiong et al., 2023), emphasize preserving data distributions through synthetic data generation. One-shot approaches, such as DOSFL (Zhou et al., 2020), FedD3 (Song et al., 2023), DENSE (Zhang et al., 2022), and Co-Boosting Dai et al. (2024), perform nearly as well as with the centralized case by distilling client data into synthetic subsets, while iterative techniques, including FedDM (Xiong et al., 2023) and FedAF (Wang et al., 2024), repeatedly refine distilled datasets to enhance training efficiency and mitigate heterogeneity. Despite these advancements, the primary use of dataset distillation is to enhance communication efficiency, without exploring its potential to eliminate error accumulation. Additionally, almost all the methods perform a single type of matching, suffering inherent limitations of that type. Our work fills these gaps by combining complementary strengths of gradient and distribution matching to unleash the full potential of dataset distillation in heterogeneous federated learning.

## 3 PRELIMINARY: FEDERATED OPTIMIZATION

FL is a privacy-preserving multi-client joint training framework involving $K$ clients (McMahan et al., 2017). Each client holds local data $\mathcal{D}_k = \{(x_k^i, y_k^i)\}_{i=1}^{n_k}$, where $n_k = |\mathcal{D}_k|$ represents

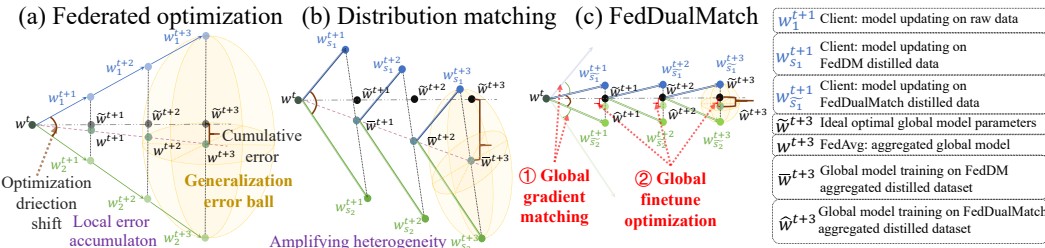

Figure 3: (a) Illustration of the optimization error accumulation in FedAvg, where the generalization error ball radius keeps growing. (b) While distribution matching distillation eliminates error accumulation, it also amplifies data heterogeneity. (c) FedDualMatch mitigates both issues by incorporating global gradient matching and global fine-tuning optimization.

the size of the dataset on client $k$ and $n = \sum_{k=1}^{K} n_k$, with $x_k^i$ and $y_k^i$, as the $i$-th sample on the $k$-th client and the corresponding label, respectively, following distribution $\mathscr{D}_k$. The goal of federated learning is to jointly train a single model parameterized by $w$ across clients, via the optimization objective is: $\min_w L(w) = \sum_{k=1}^{K} \frac{n_k}{n} L_k(w)$, where $L_k(w)$ denotes the local loss function of client $k$. The local loss function under the typical empirical risk minimization is expressed as $L_k(w) = \frac{1}{n_k} \sum_{i=1}^{n_k} \ell(f_w(x_k^i), y_k^i)$, where $\ell(\cdot)$ represents the loss function, such as cross-entropy loss or mean squared error, and $f_w(x_k^i)$ denotes the prediction of the model with parameters $w$ on input $x_k^i$. For the classic FedAvg algorithm (McMahan et al., 2017), the server first distributes the global model to each client. Each client then performs local updates on their local datasets. Specifically, the local model update for client $k$ is computed as $w_k^{t+1} = w^t - \eta \nabla L_k(w^t) = w^t - \frac{\eta}{n_k} \sum_{i=1}^{n_k} \nabla_{w^t} \ell(f_{w^t}(x_k^i), y_k^i)$, where $\eta$ is the local learning rate. After local updates, clients send their updated model parameters back to the server, where they are aggregated as $w^{t+1} = \sum_{k=1}^{K} \frac{n_k}{n} w_k^{t+1}$.

In an ideal scenario where the global aggregation is performed immediately after each local gradient update, the resulting gradient optimization would be equivalent to that of training directly on the centralized aggregated data. Let $\mathcal{D} = \bigcup_{k=1}^{K} \mathcal{D}_k = \{(x_i, y_i)\}_{i=1}^{n}$. Then the gradient updating in this case can be written as:

$$w^{t+1} = w^t - \eta \sum_{k=1}^{K} \frac{1}{n} \sum_{i=1}^{n_k} \nabla_{w^t} \ell(f_{w^t}(x_k^i), y_k^i) = w^t - \frac{\eta}{n} \sum_{i=1}^{n} \nabla_{w^t} \ell(f_{w^t}(x_i), y_i) = w^t - \eta \nabla L(w^t).$$

Let $\tilde{w}^{t+1} := w^t - \eta \nabla L(w^t), \tilde{w}^{t+E} := w^t - \eta \sum_{t'=t}^{t'=t+E-1} \nabla L(\tilde{w}^{t'})$ represents parameter updating under ideal centralized training. However, given the significant communication overhead incurred in this case, practical implementations typically defer the global aggregation until after $E$ local updates on each client, which is presented as $w_k^{t+E} := w^t - \eta \sum_{t'=t}^{t'=t+E-1} \nabla L_k(w_k^{t'}), w^{t+E} = \frac{1}{n} \sum_{k=1}^{K} \frac{n_k}{n} w_k^{t+E}$. In the common case of the inherent heterogeneity $\mathscr{D}_i \neq \mathscr{D}_j, i \neq j$ in client data distributions, the delay can give rise to significant shifts in gradient during optimization. As illustrated in Fig. 3 (a), the shift accumulates over local updates such that the trajectory of the global model deviates from the optimal path and consequently leads to sub-optimal performance and unstable convergence. The shift accumulation can be decomposed:

$$\tilde{w}^{t+E} - w^{t+E} = \eta \sum_{t'=t}^{t'=t+E-1} \sum_{k=1}^{K} \sum_{i=1}^{n_k} \left[ \nabla_{w_k^{t'}} \ell(f_{w_k^{t'}}(x_k^i), y_k^i) - \nabla_{\tilde{w}^{t'}} \ell(f_{\tilde{w}^{t'}}(x_k^i), y_k^i) \right]. \quad (1)$$

Therefore, in order to eliminate such shifts and achieve optimal joint training, we have to tackle the following two challenges: 1) how to mitigate the error accumulation over multiple local updates in Eq.(1); 2) how to reduce the gradient deviations caused by the client data heterogeneity, as seen in Fig. 3 (a), to align gradients.

## 4 METHODOLOGY: FEDDUALMATCH

To address the aforementioned challenges, we propose a federated learning framework based on dataset distillation and built upon a dual matching approach, dubbed as FedDualMatch. It performs

local distribution matching to leverage client data features and global gradient matching to correct the gradient shift on the server. By alternating the local and global optimization, we can make the most of the complementarity between the two matching distillations.

## 4.1 LOCAL DISTRIBUTION MATCHING (LDM)

To effectively extract data features while maintaining their distributional diversity on each client, we adopt a local distribution matching strategy for dataset distillation similar to FedDM(Xiong et al., 2023). For client $k$ with data $\mathcal{D}_k$, a common way is to estimate the data distribution deviation in a lower-dimensional feature space using maximum mean discrepancy (MMD) (Gretton et al., 2012):

$$\underset{\mathcal{S}_k}{\operatorname{argmin}}\left\|\mathbb{E}[h_w(\mathcal{D}_k)] - \mathbb{E}[h_w(\mathcal{S}_k)]\right\|,$$

where $h_w$ is the embedding function that maps the input data to the feature space, and $\mathcal{S}_k$ represents the distilled dataset of client $k$. To fully leverage the utility of local data in training, the choice of $h_w$ plays a crucial role. Different from FedDM Xiong et al. (2023), we introduce the concept of *generalization error ball with adaptive radius $R_t$*, for which we define $R_t$ as the largest norm of the model updating parameter difference between locally distilled dataset on any client, say $\mathcal{S}_k$, and the aggregated one $\mathcal{S} = \{\mathcal{S}_k\}_{k=1}^K$ on the server side:

$$R_t = \sup_k \left\|w_t^{\mathcal{S}_k} - w_t^{\mathcal{S}}\right\|, \tag{2}$$

where $w_t^{\mathcal{S}_k}$ and $w_t^{\mathcal{S}}$ represent the updates after one gradient step on distilled datasets $\mathcal{S}_k$ on client $k$ and $\mathcal{S}$ on the server, respectively, i.e., $w_t^{\mathcal{S}_k} = w_{t-1} - \eta\nabla L(w_{t-1}; \mathcal{S}_k)$ and $w_t^{\mathcal{S}} = w_{t-1} - \eta\nabla L(w_{t-1}; \mathcal{S})$ with $w_{t-1}$ being the model parameter from the last communication round. $R_t$ is computed on the server. The generalization error ball then refers to the one with the downloaded model parameters from the server as its center and $R_t$ as the radius, denoted by $\mathcal{B}(w_t; R_t)$ and visualized in Fig. 3. We set embedding function $h_w$ to be part of the randomly sampled model parameter $w$ from the ball $\mathcal{B}(w_t; R_t)$, and update synthetic data $S_k$ to perform feature distribution matching with client data $D_k$ in embedding space $h_w(*)$:

$$\underset{\mathcal{S}_k}{\operatorname{argmin}} \sup_{w \in \mathcal{B}(w_t; R_t)} \left\|\mathbb{E}[h_w(\mathcal{D}_k)] - \mathbb{E}[h_w(\mathcal{S}_k)]\right\|.$$

In practice, we do empirical estimation in MMD to fit the data distribution for each class:

$$L_{MMD}^k(\mathcal{S}_k; \mathcal{D}_k) = \sum_{c=1}^{C}\left\|\frac{1}{n_k^c}\sum_{i=1}^{n_k^c} h_w(x_k^{c,i}) - \frac{1}{s_k^c}\sum_{i=1}^{s_k^c} h_w(\tilde{x}_k^{c,i})\right\|,$$

where $x_k^{c,i} \in \mathcal{D}_k$ and $\tilde{x}_k^{c,i} \in \mathcal{S}_k$ represent the $i$-th sample of class $c$ in $\mathcal{D}_k$ and $\mathcal{S}_k$, respectively, while $n_k^c$ and $s_k^c$ stand for the number of samples of class $c$ in $\mathcal{D}_k$ and $\mathcal{S}_k$, respectively, and $C$ is the total number of classes.

To fully exploit the feature distribution of data on each client, in addition to the dynamic computation of the generalization error ball's radius, we put forward the *backward layer-wise feature alignment*, and incorporate it into the training process. Embedding function $h_w$ has multiple layers, i.e., $h_w = \{v_1, v_2, \ldots, v_J\}$, with $v_p$ being the $p$-th layer and $J$ being the layer number. We align each layer's feature distributions sequentially, starting from the last layer, i.e., the logit layer. Experiments show that directly summing the MMD losses of all layers for parameter updates causes convergence issues, which may result from varying complexities of the feature alignment across layers. Besides, we observed that aligning deeper (output) layers first can facilitate the learning of earlier (input) layers. Thus, to do the backward feature alignment for each layer on top of the already aligned deeper layers, our strategy is to accumulate the MMD losses of the current layer, say $p \in \{J, J-1, \ldots, 1\}$, and all deeper layers as follows:

$$L_{DM}^{k,p}(\mathcal{S}_k; \mathcal{D}_k) = \sum_{q=p}^{J} L_{MMD}^{k,v_q}(\mathcal{S}_k; \mathcal{D}_k). \tag{3}$$

This strategy enables feature distributions of deeper layers to remain aligned while aligning the feature distribution of the current layer, thus effectively extracting the information on the feature distribution of data on each client.

---

**Algorithm 1:** FedDualMatch: Dual Matching Federated Learning

---

**Input:** Random initial global model parameter $w_0$, initial radius of generalization error ball $R_0 = 5.0$, # clients $K$, communication rounds $T$, # epochs of global model training $T_g$, # local epochs $E$, learning rate $\eta$, local datasets $\{\mathcal{D}_k\}$, # layers in embedding function $J$, # rounds of global gradient matching $M$ # $\sigma$ is one constant factor on the Gaussian noise variance, # gradient norm bound $C$

**On server:**

    **for** *each communication round $t = 1, 2, \ldots, T$* **do**

        Broadcast global model parameter $w_{t-1}$ and radius $R_{t-1}$ to all clients

        Aggregate distilled datasets $\mathcal{S} = \{\mathcal{S}_k\}_{k=1}^K$ from all clients

        Update radius of error ball: $R_t = \sup_k \left\| w_t^{\mathcal{S}_k} - w_t^{\mathcal{S}} \right\|$

        **for** *each GGM round $m = 1, 2, \ldots, M$* **do**

            Randomly sample model parameter $w_g$ from error ball $\mathcal{B}(w_{t-1}; R_t)$

            Synthesize $\mathcal{S}' = \{\mathcal{S}'_k\}$ by minimizing global gradient matching loss with initial $\mathcal{S}'_k = \mathcal{S}_k$:

            $L_{GGM}(\mathcal{S}'; \mathcal{S}) = \sum_{k=1}^K \mathrm{dist}\left(\bigtriangledown_{w_g} L_k(w_g; \mathcal{S}'_k), \bigtriangledown_{w_g} L_k(w_g; \mathcal{S})\right)$

        **end**

        Fine-tune global model $w_{t-1}$ on $\tilde{\mathcal{S}} = \bigcup_{k=1}^K \{\mathcal{S}'_k, \mathcal{S}_k\}$ for $T_g$ epochs to get $w_t$

    **end**

**On each client:**

    Randomly sample model parameter $w^k$ from error ball $\mathcal{B}(w_{t-1}; R_t)$

    Use the first $J$ layers in model with parameter $w_{t-1}$ as embedding function $h_w$

    Initialize $\mathcal{S}_k$ with random Gaussian noise

    **for** *each model layer $p = J, J-1, \ldots, 1$* **do**

        Distill local dataset $\mathcal{D}_k$ for synthetic dataset $\mathcal{S}_k$ via minimizing distribution matching loss:

        $L_{DM}^{k,p}(\mathcal{S}_k; \mathcal{D}_k) = \sum_{q=p}^J L_{MMD}^{k, v_q}(\mathcal{S}_k; \mathcal{D}_k)$

        Adding Gaussian noise-based differential privacy:

        Obtain the clipped gradient:   $\nabla_{\mathcal{S}_k} L_{DM}^{k,p}(\mathcal{S}_k; \mathcal{D}_k) \leftarrow \dfrac{\nabla_{\mathcal{S}_k} L_{DM}^{k,p}(\mathcal{S}_k; \mathcal{D}_k)}{\max\left(1, \frac{\|\nabla_{\mathcal{S}_k} L_{DM}^{k,p}(\mathcal{S}_k; \mathcal{D}_k)\|_2}{C}\right)}$

        Add Gaussian noise:   $\nabla_{\mathcal{S}_k} L_{DM}^{k,p}(\mathcal{S}_k; \mathcal{D}_k) \leftarrow \nabla_{\mathcal{S}_k} L_{DM}^{k,p}(\mathcal{S}_k; \mathcal{D}_k) + \dfrac{1}{|\mathcal{S}^k|} \mathcal{N}(0, \sigma^2 C^2 \mathbf{I})$

    **end**

    Upload distilled dataset $\mathcal{S}_k$ to server

---

## 4.2 GLOBAL GRADIENT MATCHING (GGM)

On the server side, the global model is trained on the aggregated distilled dataset in the same manner as it would be on a centralized dataset, thus avoiding the issue of local error accumulation over multiple local updates depicted in Fig. 3 (a). Specifically, after sufficient LDM distillation on $K$ clients, distilled dataset on each client is sent to the server and aggregated: $\mathcal{S} = \{\mathcal{S}_k\}_{k=1}^K$. However, LDM-based dataset distillation exacerbates the data heterogeneity across clients, as demonstrated in Huang et al. (2023) and presented in Fig. 3 (b), such that the optimization of the model trained on each client distilled data $\mathcal{S}_k$ may diverge from the optimal path and thus lead to sub-optimal convergence. To address this issue, we perform two operations, global gradient matching (GGM) and global fine-tuning optimization (GFO), to improve federated optimization stability and model performance, as illustrated in Fig. 3 (c).

Global gradient matching for dataset distillation aims to align gradients on locally distilled data from each client $\mathcal{S}_k$ and that on the aggregated one $\mathcal{S}$ by minimizing the following loss with initial $\mathcal{S}'_k$ set to $\mathcal{S}_k$:

$$L_{GGM}(\mathcal{S}'; \mathcal{S}) = \mathop{\mathbb{E}}_{w \in \mathcal{B}(w_{t-1}; R_t)} \left[\sum_{k=1}^K \mathrm{dist}\left(\bigtriangledown_w L_k(w; \mathcal{S}'_k), \bigtriangledown_w L_k(w; \mathcal{S})\right)\right], \qquad (4)$$

where $\mathcal{S}' = \{\mathcal{S}'_k\}$, and $\mathrm{dist}$ computes the distance between the above two types of gradient for which we choose the same as in Zhao et al. (2020) to ensure that gradients are aligned.

To preserve data diversity, we further incorporate the newly distilled dataset $\mathcal{S}'_k$ into $\mathcal{S}_k$ and fine-tune the global model by running gradient steps on the new aggregated data $\tilde{\mathcal{S}} = \bigcup_{k=1}^K \{\mathcal{S}'_k, \mathcal{S}_k\}$ for

$T_g$ epochs[1], which ensures that the global model benefits from both aligned gradients and diverse data representations. To promote the effect of the above two operations on the global model, we perform them alternately on the server side for $M$ rounds. In each round, the global model is randomly sampled from the generalization error ball of adaptive radius $R_t$ to enhance the robustness of gradient matching. The interplay between gradient alignment and fine-tuning helps stabilize the convergence and improve performance.

The whole algorithm of FedDualMatch is summarized in Algorithm 1.

## 5 THEORETICAL ANALYSIS

We now conduct a theoretical analysis for our algorithm to reveal its properties on effectiveness, convergence, and privacy. While prior work DynaFed (Pi et al., 2023) has explored the convergence of distilled data fine-tuning as a substitute for model communication, the assumptions are too strong and the theoretical justification remains limited. To see why it is effective to replace model with distilled dataset for communication in FL, we begin by introducing the following assumptions:

**Assumption 1** (Smooth and convex). *Local objective function $L_k(\cdot)$ and embedding functions $h_w(\cdot)$ for any client $k = 1, \cdots, K$ are $L$-smooth and $\mu$-strongly convex.*

**Assumption 2** (Bounded gradient). *For each client, the norm of the gradient is bounded:*

$$\|\nabla L_k(w)\| \leq G, \quad \forall w \in \mathbb{R}^d, \quad \forall k \in \{1, 2, \ldots, K\}, \tag{5}$$

*where $G$ is a positive constant.*

**Assumption 3** (Bounded distributional distance). *For any embedding model with parameters randomly sampled from a generalization error ball centered at $w^t$ with radius $R$, the distilled dataset $\mathcal{S}_k$ and the given dataset $\mathcal{D}_k$ on the $k$-th client are mapped to the feature space by the embedding model. The expected feature distance between them is bounded by a small constant $\sigma_d > 0$ for any sampled $w$:*

$$\left\|\mathbb{E}[h_w(\mathcal{D}_k)] - \mathbb{E}[h_w(\mathcal{S}_k)]\right\| \leq \sigma_d, \quad \forall w \in \mathcal{B}(w^t, R), \quad \forall k \in \{1, 2, \ldots, K\}. \tag{6}$$

**Assumption 4** (Bounded gradient difference). *The difference between the gradients computed on the distilled dataset $\mathcal{S}_k$ and the aggregated one $\mathcal{S}$ is bounded by a constant $\sigma_g > 0$ in expectation for any $w$ in the training trajectory of global model parameter:*

$$\mathbb{E}\left[\left\|\nabla_w L(w, \mathcal{S}_k) - \nabla_w L(w, \mathcal{S})\right\|\right] \leq \sigma_g, \ \forall w \in \{w^t | w^{t+1} = w^t - \eta \nabla L(w, \mathcal{S}), t=0, ..., T_g-1\}. \tag{7}$$

Assumption 1 ensures that local optimization problems are well-behaved, with smooth and convex functions, for global convergence. Assumption 2 ensures that gradients remain controlled. Assumption 3 bounds the difference in data distribution between the distilled and original datasets to ensure that the distillation process won't introduce significant distortions. Assumption 4 ensures that gradients computed on distilled datasets closely approximate those computed on the aggregated dataset for minimizing error accumulation. Under Assumptions 1-4, we have the following theorem:

**Theorem 1** (Effectiveness). *Let $L(w_s^*; \mathcal{D})$ denote the loss of the model with optimal parameter $w_s^*$ trained on the aggregated distilled dataset $\mathcal{S}$, evaluated on the original dataset $\mathcal{D}$, and $w_d^*$ present the optimal parameter trained on $\mathcal{D}$. With learning rate $\eta = \frac{c}{T_g}$ for $c \geq \frac{G}{\mu \sigma_g}$ and distributional distance bound $\sigma_d = \sqrt{2\mu^2 \epsilon_c / L}$ for small positive constant $\epsilon_c$, it holds that:*

$$\mathbb{E}\left[\left\|L(w_s^*; \mathcal{D}) - L(w_d^*; \mathcal{D})\right\|\right] \leq \varepsilon, \tag{8}$$

*where $\varepsilon = \frac{L\sigma_d^2}{2\mu^2} \leq \epsilon_c$ is a sufficiently small positive constant.*

Theorem 1 demonstrates that the loss of the model trained on distilled data closely approximates the loss of the model trained on original data, which proves that aggregated distilled dataset training can achieve a similar training effect to centralized training. This supports the feasibility of using distilled data as an alternative for federated communication.

Regarding convergence, the following theorem holds.

---

[1] Global model in fine-tuning is indexed by $w_{t-1,s}$ for epoch $s = 0, 1, \cdots, T_g - 1$, with $w_{t-1,0} = w_{t-1}$ and $w_t := w_{t-1, T_g-1}$.

**Theorem 2** (Convergence). *Define $C = \frac{T_g \eta L^2 (L\sigma_d^2 + \eta G^2)}{2(T_g \mu^2 \eta - L)}$. Under the same assumptions as in Theorem 1, it holds that:*

$$E\left[L(w_t; \mathcal{D})\right] \le L(w_d^*; \mathcal{D}) + C, \tag{9}$$

*Remark* (Convergence per communication round). Define $C_t = \frac{L \cdot E[L(w_{t-1}; \mathcal{D})]}{\mu^2 \eta}$ and $B = \frac{L^2 (L\sigma_d^2 + \eta G^2)}{2\mu^2}$. Under the same assumptions as in Theorem 1, it holds that

$$E\left[L(w_t; \mathcal{D})\right] - L(w_d^*; \mathcal{D}) \le \frac{C_t}{T_g} + B, \tag{10}$$

where $T_g$ is the number of training epochs of the global model on the aggregated distilled dataset in a single communication round.

Theorem 2 shows that our algorithm converges with the sub-optimality gap of the final model bounded by a constant determined by the distributional distance $\sigma_d$. Combined with Theorem 1, this indicates that the effectiveness and convergence of FedDualMatch depend on the accuracy of local client data distillation in matching the data distribution. Remark 5 further highlights that during a single communication round, the server-side model can achieve strong convergence on the distilled data. This is because a large number of global training iterations $T_g$ can be performed within a single round, avoiding error accumulation in federated optimization. Previous work, such as DynaFed (Pi et al., 2023), provides convergence guarantees for federated learning with distilled data. However, their analysis assumes that gradients on distilled and original data are tightly aligned across the entire parameter space, formalized as $\|\nabla L(w; \mathcal{S}) - \nabla L(w; \mathcal{D})\| \le \delta \|\nabla L(w; \mathcal{D})\| + \epsilon$ for all $w$. This assumption is overly restrictive and unrealistic, as perfect gradient alignment across the full parameter space is nearly unattainable. To relax this limitation, we confine the model parameters to a localized ball, as stated in Assumption 3. This weaker assumption is more practical and can be satisfied by performing sufficient gradient matching through random sampling within the ball. This refinement not only broadens the applicability of our approach but also emphasizes its practical convergence in federated learning scenarios.

In addition, to ensure differential privacy during the execution of our algorithm, we adopt the Gaussian-based differential privacy mechanism, following the approach used in FedDM. Compared to FedDM, our method introduces an additional step of sharing the Gaussian error ball radius $R_t$ between the client and server. However, $R_t$ is privacy-insensitive as it is independent of client data. This ensures that FedDualMatch also achieves an $(\epsilon, \delta)$-differential privacy guarantee. The detailed proofs of these theorems are provided in Appendices A and B.

## 6 EXPERIMENTS

### 6.1 EXPERIMENTAL SETUP

**Datasets.** We evaluate FedDualMatch through controlled experiments and naturally heterogeneous data. In controlled experiments, client heterogeneity is adjusted using the Dirichlet distribution's alpha parameter, where smaller values indicate greater heterogeneity (McMahan et al., 2017). The MNIST (LeCun et al., 1998) and CIFAR10 (Krizhevsky et al., 2009) datasets are distributed to 10 clients with alpha values of 0.005, 0.01, and 0.05, respectively, to test the performance under varying degree of disparities in data distribution. For real-world heterogeneity, the Digital-Five dataset—comprising MNIST (LeCun et al., 1998), MNIST-M (Ganin et al., 2016), SVHN (Netzer et al., 2011), SynthDigits (Hull, 1994), and USPS (Ganin & Lempitsky, 2015)—is naturally divided into 5 datasets, one for each client, exhibiting significant differences in image style and source. Similarly, Office-Home (Venkateswara et al., 2017) is split into four clients: Art, Clipart, Product, and Real World, each in a different visual domain.

**Baseline methods.** We compare FedDualMatch with classical federated learning methods performing model communication, including FedAvg (McMahan et al., 2017), FedProx (Li et al., 2020), FedNova (Wang et al., 2020b), and SCAFFOLD (Karimireddy et al., 2020). The latter three specialize in addressing client data heterogeneity. We also compare FedDualMatch with methods doing distilled dataset communication, such as FedDM (Xiong et al., 2023) and DynaFed (Pi et al., 2023).

| Methods | MNIST | | | CIFAR10 | | |
|---|---|---|---|---|---|---|
| | $\alpha = 0.05$ | $\alpha = 0.01$ | $\alpha = 0.005$ | $\alpha = 0.05$ | $\alpha = 0.01$ | $\alpha = 0.005$ |
| FedAvg | 97.93±0.02 | 80.09±1.57 | 82.95±1.69 | 53.64±3.39 | 39.46±0.88 | 38.04±2.92 |
| FedProx | 98.01±0.00 | 89.99±0.98 | 83.41±2.34 | 55.13±0.04 | 40.93±1.68 | 39.30±2.02 |
| FedNova | 97.67±0.01 | 75.68±0.14 | 77.97±5.35 | 47.90±2.65 | 37.89±0.74 | 25.97±3.32 |
| SCAFFOLD | 97.94±0.03 | 86.06±0.93 | 79.99±2.50 | 49.62±1.17 | 17.14±2.11 | 33.83±0.13 |
| DYNEFED | **98.13±0.04** | 95.69±0.93 | 75.50±6.28 | **57.23±1.56** | 43.06±3.61 | 27.41±1.94 |
| FedDM | 95.43±0.02 | 95.38±0.12 | 95.52±0.03 | 42.44±0.15 | 41.15±0.12 | 41.99±0.19 |
| FedDualMatch | 96.94±0.03 | **97.03±0.04** | **97.00±1.11** | 44.61±1.02 | **44.98±0.40** | **43.75±0.68** |

Table 1: Accuracy of FL algorithms on MNIST and CIFAR10 in controlled experiments. The results demonstrate that FedDualMatch maintains accuracy for increasing data heterogeneity between clients (smaller $\alpha$).

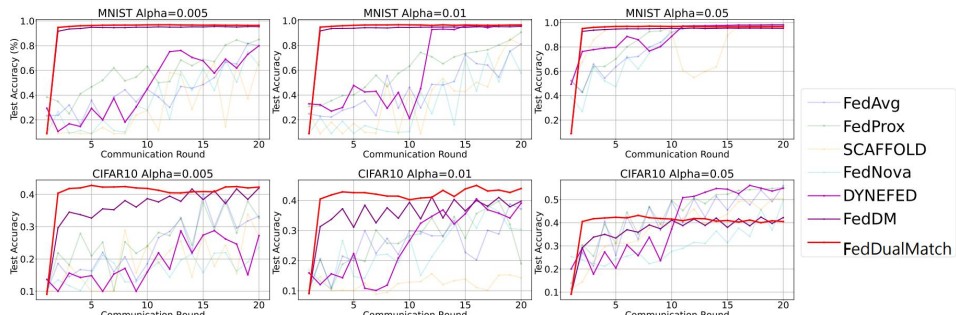

Figure 4: Test accuracy under different communication rounds. FedDualMatch's accuracy increases rapidly in the first round of communication.

Particularly, DynaFed leverages both types of communication by running FedAvg in the early stages and then fine-tuning on the aggregated distilled dataset.

**Hyperparameters.** In controlled experiments, all clients participate in $T = 20$ communication rounds with batch size 256, which is 10 or 64 for naturally heterogeneous data. Local distribution matching runs 200 iterations with learning rate $\eta = 1.0$, while global gradient matching runs $M = 10$ rounds of sampling model $w_g$ and synthesizing data $\mathcal{S}'$ by running 10 iterations with learning rate $\eta = 0.1$. Fine-tuning global model runs $T_g = 500$ iterations with learning rate $\eta_m = 0.001$ per communication round. The distilled data are initialized by random Gaussian noise to safeguard local data privacy. In addition, only the pooling layers are selected for layer-wise alignment and the layer number $J = 3$. If there are no special instructions, we use the Gaussian noise with $\sigma_r = 0, \sigma_s = 0$.

## 6.2 RESULTS AND ANALYSIS

**Effectiveness against heterogeneity.** Experimental results on heterogeneity are reported in Tab. 1 and Tab. 2. Results from controlled experiments in Tab. 1 indicate that increasing the degree of client heterogeneity ($\alpha$ decreasing from 0.05 to 0.005) makes model-communication-based federated learning methods significantly less effective. For instance, SCAFFOLD's accuracy drops from 97.94% to 79.99% on MNIST, and from 49.62% to 33.83% on CIFAR-10. In contrast, FedDualMatch performs well and outperforms FedDM, consistently across different degrees of heterogeneity. Particularly, it outperforms all the baselines for the case of high degree of heterogeneity, e.g., $\alpha = 0.05$ or 0.005, and its advantage over baselines becomes more pronounced for the highest degree of heterogeneity, i.e., $\alpha = 0.005$. However, in scenarios of low degree of heterogeneity, the performance of dataset distillation methods remains limited, due to the small number of images per class and thus potential overfitting during training in this case. We leave it to our future work to address the challenge of how to make the data distillation best possible for FL across the full spectrum of heterogeneity. Moreover, experimental results in Tab. 2 further confirm FedDualMatch's effectiveness in real-world heterogeneous scenarios. For example, FedDualMatch achieves relative improvements of 24.56% on Digital-Five and 17.22% on Office-Home compared to FedDM, especially achieving an absolute improvement of 27.37% on SVHN.

**Convergence.** Fig. 4 is about the convergence behaviors of FL algorithms. FedDualMatch, thanks to GGM, converges significantly faster than FedDM, particularly on CIFAR10 where it outperforms all other baselines in a single communication round.

Figure 5: Comparisons of privacy, ablation study, and different image per class (IPC).

| Methods | Digital-Five | | | | | Office-Home | | | |
|---|---|---|---|---|---|---|---|---|---|
| | MNIST | M-M | SVHN | SYN | USPS | Art | Clipart | Product | Real |
| FedAvg | 91.86 | 69.78 | 49.63 | 61.01 | 92.04 | 9.52 | 21.55 | 27.43 | 13.81 |
| FedProx | 92.31 | 75.78 | 50.12 | 62.01 | 93.17 | 9.26 | 22.16 | 26.42 | 13.48 |
| FedNova | 91.99 | 76.12 | 43.25 | 62.37 | 93.06 | 9.26 | 20.53 | 26.97 | 13.20 |
| SCAFFOLD | 91.41 | 73.07 | 26.76 | 51.83 | 90.97 | 7.31 | 16.16 | 20.50 | 11.38 |
| FedDM | 90.71 | 64.80 | 34.93 | 60.90 | **93.55** | 11.47 | 21.64 | 29.45 | 19.48 |
| FedDualMatch | **94.57** | **77.13** | **62.30** | **75.20** | 91.40 | **13.00** | **25.08** | **35.05** | **23.50** |

Table 2: Accuracy of FL algorithms on Digital-Five and Office-Home. FedDualMatch can resist the natural heterogeneity of data and excel in knowledge discovery across clients.

**Privacy.** Fig. 5 (a) shows the privacy impact of Gaussian noise. Increasing $\sigma_s$ in the distilled data leads to a decline in performance. The noise effect on the dynamic radius $\sigma_r$ gets more pronounced with larger $\sigma_s$, though whether this is positive or negative depends on the trade-off between stability and convergence. Smaller radii may improve stability due to the reduced sample space, but leave optimal model parameters out of the error ball. A larger radius may cover optimal model parameters, but reduce stability due to a large sample space. How to better balance stability and performance by optimizing $\sigma_r$ is also an interesting future work.

**Impact of IPC.** Fig. 5 (c) indicates that both small and large sizes of distilled dataset (IPC) decrease performance, which is consistent with observations in previous data distillation research (Lee & Chung, 2024; Guo et al., 2023).

**Ablation study.** Ablation studies in Fig. 5 (b) demonstrate that the layer-wise feature alignment in LDM brings a significant performance boost, and GGM can further improve performance (e.g., 0.42% on MNIST and 0.93% on CIFAR10), validating the importance of the combination.

## 7 CONCLUSION

We propose a federated learning framework, termed *FedDualMatch*, that leverages distilled data for communication to eliminate the error accumulation caused by data heterogeneity across clients. It's characterized by dual matching, i.e., distribution matching on clients and gradient matching on server. Moreover, we introduce layer-wise feature alignment for distribution matching and global model fine-tuning for gradient matching. By design, FedDualMatch excels in facilitating distribution knowledge extraction and convergence stability. We further conduct a theoretical analysis to understand the rationale behind the practicality of communicating distilled data in federated learning in terms of effectiveness, convergence, and privacy. Extensive experiments in both controlled and real-world heterogeneity settings demonstrate its superior and stable performance, and prominent advantages over the state-of-the-arts on highly heterogeneous data.

**Limitation:** The performance of FedDualMatch shows stagnation in low-heterogeneity scenarios, indicating space for improvement. Additionally, current data distillation techniques require substantial computing resources, which may be beyond the computational capacity of edge devices.

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

## A  PROOF OF EFFECTIVENESS

**Theorem 3** (Theorem of effectiveness restated). *Let $L(w_s^*; \mathcal{D})$ denote the loss of the model with optimal parameter $w_s^*$ trained on the aggregated distilled dataset $\mathcal{S}$, evaluated on the original dataset $\mathcal{D}$, and $w_d^*$ the optimal parameter trained on $\mathcal{D}$. With learning rate $\eta = \frac{c}{T_g}$ for $c \geq \frac{G}{\mu \sigma_g}$ and distributional distance bound $\sigma_d = \sqrt{2\mu^2 \epsilon_c / L}$ for small positive constant $\epsilon_c$, it holds under Assumptions 1-4 that:*

$$\mathbb{E}\left[\left\|L(w_s^*; \mathcal{D}) - L(w_d^*; \mathcal{D})\right\|\right] \leq \varepsilon, \tag{11}$$

*where $\varepsilon = \frac{L\sigma_d^2}{2\mu^2} \leq \epsilon_c$ is a sufficiently small positive constant.*

*Proof.* The overall objective function $L(\cdot)$ is $L$-smooth and $\mu$-strongly convex, i.e.,

$$L(w'; \mathcal{D}) \geq L(w; \mathcal{D}) + \nabla L(w; \mathcal{D})^T (w' - w) + \frac{\mu}{2} \|w' - w\|^2 \tag{12}$$

and

$$L(w'; \mathcal{D}) \leq L(w; \mathcal{D}) + \nabla L(w; \mathcal{D})^T (w' - w) + \frac{L}{2} \|w' - w\|^2. \tag{13}$$

Since $w_s^*$ is the optimal parameter trained on the aggregated distilled dataset $\mathcal{S}$ and $w_d^*$ is the optimal parameter trained directly on $\mathcal{D}$, it holds that

$$\nabla L(w_d^*; \mathcal{D}) = 0$$

and

$$\nabla L(w_s^*; \mathcal{S}) = 0. \tag{14}$$

Substituting $w' = w_s^*$ and $w = w_d^*$ into Eq.(12) and Eq.(13), we obtain:

$$L(w_s^*; \mathcal{D}) \geq L(w_d^*; \mathcal{D}) + \frac{\mu}{2} \|w_s^* - w_d^*\|^2 \tag{15}$$

and

$$L(w_s^*; \mathcal{D}) \leq L(w_d^*; \mathcal{D}) + \frac{L}{2} \|w_s^* - w_d^*\|^2 \tag{16}$$

By Eq. 15 and Eq.(16), we obtain

$$|L(w_s^*; \mathcal{D}) - L(w_d^*; \mathcal{D})| \leq \frac{L}{2} \|w_s^* - w_d^*\|^2.$$

Since $L(\cdot)$ is $\mu$-strongly convex, we can use the following gradient-based characterization:

$$\langle \nabla L(x) - \nabla L(y),\ x - y \rangle \geq \mu \|x - y\|^2$$

for all $x, y \in \mathbb{R}^n$. Applying this to $w_t$ and $w_s^*$, we have:

$$\langle \nabla L(w_t; \mathcal{S}) - \nabla L(w_s^*; \mathcal{S}),\ w_t - w_s^* \rangle \geq \mu \|w_t - w_s^*\|^2. \tag{17}$$

Next, applying the Cauchy-Schwarz inequality to the left-hand side of Eq.(17), we obtain:

$$\|\nabla L(w_t; \mathcal{S}) - \nabla L(w_s^*; \mathcal{S})\| \, \|w_t - w_s^*\| \geq \langle \nabla L(w_t; \mathcal{S}) - \nabla L(w_s^*; \mathcal{S}), \, w_t - w_s^* \rangle \geq \mu \|w_t - w_s^*\|^2$$

Assuming $w_t \neq w_s^*$, we can divide both sides by $\|w_t - w_s^*\|$ to obtain:

$$\|\nabla L(w_t; \mathcal{S}) - \nabla L(w_s^*; \mathcal{S})\| \geq \mu \|w_t - w_s^*\|. \tag{18}$$

From Eq.(18), we can bound the parameter distance by Assumption 2:

$$\|w_t - w_s^*\| \leq \frac{\|\nabla L(w_t; \mathcal{S})\|}{\mu} \leq \frac{G}{\mu}. \tag{19}$$

To constrain the parameter distance within the dynamic generalization error ball, we define:

$$R_t = \sup_k \left\| w_t^{\mathcal{S}_k} - w_t^{\mathcal{S}} \right\|.$$

Using the parameter update rules:

$$w_t^{\mathcal{S}_k} = w_{t-1}^{\mathcal{S}} - \eta \nabla L(w_{t-1}; \mathcal{S}_k), \quad w_t^{\mathcal{S}} = w_{t-1}^{\mathcal{S}} - \eta \nabla L(w_{t-1}; \mathcal{S}),$$

we have:

$$R_t = \eta \cdot \sup_k \|\nabla L(w_{t-1}; \mathcal{S}_k) - \nabla L(w_{t-1}; \mathcal{S})\| \leq \eta \sigma_d. \tag{20}$$

To ensure that the parameter distance $\|w_t - w_s^*\|$ is within the dynamic generalization error ball, we combine Eq.(19) with Eq.(20), and set $\eta = \frac{c}{T_g}$:

$$\|w_t - w_s^*\| \leq \frac{G}{\mu} \leq R_t \leq c\sigma_d \quad \Longrightarrow \quad c \geq \frac{G}{\mu \sigma_d}.$$

Next, define $h_{w_s^*}(*)$ in Assumption 3 as:

$$h_{w_s^*}(*) = \nabla L(w_s^*; *).$$

The gradients of $w_s^*$ on the aggregated input dataset $\mathcal{D}$ and distilled dataset $\mathcal{S}$ can be expressed as:

$$\nabla L(w_s^*; \mathcal{D}) = \frac{1}{K} \sum_{k=1}^{K} \nabla L_k(w_s^*; \mathcal{D}_k), \quad \nabla L(w_s^*; \mathcal{S}) = \frac{1}{K} \sum_{k=1}^{K} \nabla L_k(w_s^*; \mathcal{S}_k).$$

Then, applying Assumption 3, we have:

$$\|\nabla L(w_s^*; \mathcal{D}) - \nabla L(w_s^*; \mathcal{S})\| = \left\| \frac{1}{K} \sum_{k=1}^{K} \nabla L_k(w_s^*; \mathcal{D}_k) - \frac{1}{K} \sum_{k=1}^{K} \nabla L_k(w_s^*; \mathcal{S}_k) \right\|$$

$$\leq \frac{1}{K} \sum_{k=1}^{K} \|\nabla L_k(w_s^*; \mathcal{D}_k) - \nabla L_k(w_s^*; \mathcal{S}_k)\|$$

$$\leq \sigma_d.$$

Then, applying Eq.(14) yields that

$$\|\nabla L(w_s^*; \mathcal{D})\| = \|\nabla L(w_s^*; \mathcal{D}) - \nabla L(w_s^*; \mathcal{S})\| \leq \sigma_d.$$

Applying $\mu$-strong convexity of $L(\cdot)$, Cauchy-Schwarz inequality, and Assumption 3, similar to Eq.(19), we get that

$$\|w_s^* - w_d^*\| \leq \frac{\|\nabla L(w_s^*; \mathcal{D}) - \nabla L(w_d^*; \mathcal{D})\|}{\mu} \leq \frac{\sigma_d}{\mu}.$$

Finally, using Eq.(17), the effectiveness theorem can be proven as:

$$\mathbb{E}\left[\|L(w_s^*; \mathcal{D}) - L(w_d^*; \mathcal{D})|\|\right] \leq \mathbb{E}\left[\frac{L}{2} \|w_s^* - w_d^*\|^2\right] \leq \frac{L\sigma_d^2}{2\mu^2}.$$

Since $\sigma_d = \sqrt{2\mu^2 \epsilon_c / L}$ for small positive constant $\epsilon_c$, it holds that

$$\mathbb{E}\left[\|L(w_s^*; \mathcal{D}) - L(w_d^*; \mathcal{D})|\|\right] \leq \epsilon_c.$$

$\square$

## B  PROOF OF CONVERGENCE

**Theorem 4** (Convergence). *Define $C = \frac{T_g \eta L^2(L\sigma_d^2 + \eta G^2)}{2(T_g \mu^2 \eta - L)}$. Under the same assumptions as in Theorem 1, it holds that:*

$$E\left[L(w_t; \mathcal{D})\right] \leq L(w_d^*; \mathcal{D}) + C, \tag{21}$$

*Remark* (Convergence per communication round). Define $C_t = \frac{L \cdot E[L(w_{t-1}; \mathcal{D})]}{\mu^2 \eta}$ and $B = \frac{L^2(L\sigma_d^2 + \eta G^2)}{2\mu^2}$. Under the same assumptions as in Theorem 1, it holds that

$$E\left[L(w_t; \mathcal{D})\right] - L(w_d^*; \mathcal{D}) \leq \frac{C_t}{T_g} + B, \tag{22}$$

where $T_g$ is the number of training epochs of the global model on the aggregated distilled dataset in a single communication round.

*Proof.* We begin by leveraging Assumption 3:

$$\|\mathbb{E}[h_w(\mathcal{D}_k)] - \mathbb{E}[h_w(\mathcal{S}_k)]\| \leq \sigma_d.$$

Aggregating over all $K$ clients, the expectations for the aggregated input data $\mathcal{D}$ and distilled data $\mathcal{S}$ are:

$$\mathbb{E}[h_w(\mathcal{D})] = \frac{1}{K}\sum_{k=1}^K \mathbb{E}[h_w(\mathcal{D}_k)], \quad \mathbb{E}[h_w(\mathcal{S})] = \frac{1}{K}\sum_{k=1}^K \mathbb{E}[h_w(\mathcal{S}_k)].$$

Therefore, the distributional distance between aggregated datasets $\mathcal{D}$ and $\mathcal{S}$ is:

$$\begin{aligned}
\|\mathbb{E}[h_w(\mathcal{D})] - \mathbb{E}[h_w(\mathcal{S})]\| &= \left\|\frac{1}{K}\sum_{k=1}^K \mathbb{E}[h_w(\mathcal{D}_k)] - \frac{1}{K}\sum_{k=1}^K \mathbb{E}[h_w(\mathcal{S}_k)]\right\| \\
&= \frac{1}{K}\left\|\sum_{k=1}^K \mathbb{E}[h_w(\mathcal{D}_k)] - \sum_{k=1}^K \mathbb{E}[h_w(\mathcal{S}_k)]\right\| \\
&\leq \frac{1}{K}\sum_{k=1}^K \|\mathbb{E}[h_w(\mathcal{D}_k)] - \mathbb{E}[h_w(\mathcal{S}_k)]\| \\
&\leq \sigma_d.
\end{aligned}$$

Given that the embedding function $h_w(*)$ is $L_h$-smooth, we can bound the gradient difference as:

$$\|\nabla L(w_t; \mathcal{D}) - \nabla L(w_t; \mathcal{S})\| \leq L_h \|\mathbb{E}[h_w(\mathcal{D})] - \mathbb{E}[h_w(\mathcal{S})]\| \leq L_h \sigma_d.$$

Next, consider the parameter update rule:

$$\begin{aligned}
w_{t-1,0} &= w_{t-1}, \\
w_{t-1,\hat{t}+1} &= w_{t-1,\hat{t}} - \eta \nabla L(w_{t-1,\hat{t}}; \mathcal{S}), \hat{t} = \{0, 1, \cdot, T_g - 1\}, \\
w_t &= w_{t-1, T_g - 1}.
\end{aligned}$$

Utilizing the $L$-smoothness of $L(w; \mathcal{D})$, we have:

$$\begin{aligned}
L(w_{t-1,\hat{t}+1}; \mathcal{D}) &\leq L(w_{t-1,\hat{t}}; \mathcal{D}) + \left\langle \nabla L(w_{t-1,\hat{t}}; \mathcal{D}), w_{t-1,\hat{t}+1} - w_{t-1,\hat{t}} \right\rangle + \frac{L}{2}\left\|w_{t-1,\hat{t}+1} - w_{t-1,\hat{t}}\right\|^2 \\
&= L(w_{t-1,\hat{t}}; \mathcal{D}) - \eta\left\langle \nabla L(w_{t-1,\hat{t}}; \mathcal{D}), \nabla L(w_{t-1,\hat{t}}; \mathcal{S})\right\rangle + \frac{L\eta^2}{2}\left\|\nabla L(w_t; \mathcal{S})\right\|^2.
\end{aligned} \tag{23}$$

Define the gradient error as:

$$e = \nabla L(w_{t-1,\hat{t}}; \mathcal{S}) - \nabla L(w_{t-1,\hat{t}}; \mathcal{D}).$$

Then, the inner product term can be expanded as:

$$
\begin{aligned}
\left\langle \nabla L(w_{t-1,\hat{t}}; \mathcal{D}), \nabla L(w_{t-1,\hat{t}}; \mathcal{S}) \right\rangle &= \left\langle \nabla L(w_{t-1,\hat{t}}; \mathcal{D}), \nabla L(w_{t-1,\hat{t}}; \mathcal{D}) + e \right\rangle \\
&= \left\| \nabla L(w_{t-1,\hat{t}}; \mathcal{D}) \right\|^2 + \left\langle \nabla L(w_{t-1,\hat{t}}; \mathcal{D}), e \right\rangle \\
&\geq \left\| \nabla L(w_{t-1,\hat{t}}; \mathcal{D}) \right\|^2 - \left\| \nabla L(w_{t-1,\hat{t}}; \mathcal{D}) \right\| \cdot \|e\| \\
&\geq \left\| \nabla L(w_{t-1,\hat{t}}; \mathcal{D}) \right\|^2 - L_h \sigma_d \left\| \nabla L(w_{t-1,\hat{t}}; \mathcal{D}) \right\|.
\end{aligned}
\tag{24}
$$

Substituting Eq.(24) back into Eq.(23), we obtain:

$$
\mathbb{E}[L(w_{t-1,\hat{t}+1}; \mathcal{D})]
$$
$$
\leq \mathbb{E}[L(w_{t-1,\hat{t}}; \mathcal{D})] - \eta \mathbb{E}\left[\left\| \nabla L(w_{t-1,\hat{t}}; \mathcal{D}) \right\|^2\right] + \eta L_h \sigma_d \mathbb{E}\left[\left\| \nabla L(w_{t-1,\hat{t}}; \mathcal{D}) \right\|\right] + \frac{L\eta^2 G^2}{2}.
\tag{25}
$$

To bound the cross term, we apply the inequality $ab \leq \frac{a^2}{2c} + \frac{cb^2}{2}$ with $a = \eta L_h \sigma_d$ and $b = \mathbb{E}\left[\|\nabla L(w_t; \mathcal{D})\|\right]$, choosing $c = \eta$:

$$
\begin{aligned}
\eta L_h \sigma_d \mathbb{E}\left[\left\| \nabla L(w_{t-1,\hat{t}}; \mathcal{D}) \right\|\right] &\leq \frac{(\eta L_h \sigma_d)^2}{2\eta} + \frac{\eta \left(\mathbb{E}\left[\left\| \nabla L(w_{t-1,\hat{t}}; \mathcal{D}) \right\|\right]\right)^2}{2} \\
&= \frac{\eta L_h^2 \sigma_d^2}{2} + \frac{\eta \left(\mathbb{E}\left[\left\| \nabla L(w_{t-1,\hat{t}}; \mathcal{D}) \right\|\right]\right)^2}{2}.
\end{aligned}
\tag{26}
$$

Substituting Eq.(26) into Eq.(25), we obtain:

$$
\begin{aligned}
\mathbb{E}[L(w_{t-1,\hat{t}+1}; \mathcal{D})] &\leq \mathbb{E}[L(w_{t-1,\hat{t}}; \mathcal{D})] - \eta \mathbb{E}\left[\left\| \nabla L(w_{t-1,\hat{t}}; \mathcal{D}) \right\|^2\right] \\
&\quad + \frac{\eta L_h^2 \sigma_d^2}{2} + \frac{\eta \left(\mathbb{E}\left[\left\| \nabla L(w_{t-1,\hat{t}}; \mathcal{D}) \right\|\right]\right)^2}{2} + \frac{L\eta^2 G^2}{2}.
\end{aligned}
\tag{27}
$$

Applying the $\mu$-strong convexity of $L(\cdot)$, we have:

$$
\mathbb{E}[\left\| \nabla L(w_{t-1,\hat{t}}; \mathcal{D}) \right\|^2] \geq \frac{2\mu^2}{L}\left(\mathbb{E}[L(w_{t-1,\hat{t}}; \mathcal{D})] - L(w_d^*; \mathcal{D})\right) = \frac{2\mu^2}{L}\Delta_t.
\tag{28}
$$

Let us define the sub-optimality gap:

$$
\Delta_t = \mathbb{E}[L(w_{t-1,\hat{t}}; \mathcal{D})] - L(w_d^*; \mathcal{D}).
$$

Substituting Eq.(28) into Eq.(27), we obtain:

$$
\Delta_{t+1} \leq \Delta_t - 2\mu\eta\Delta_t + \frac{\eta L_h^2 \sigma_d^2}{2} + \frac{\eta(\mathbb{E}\left[\left\| \nabla L(w_{t-1,\hat{t}}; \mathcal{D}) \right\|\right])^2}{2} + \frac{L\eta^2 G^2}{2}.
\tag{29}
$$

To handle the term $\eta(\mathbb{E}[\|\nabla L(w_{t-1,\hat{t}}; \mathcal{D})\|])^2$, we apply Jensen's inequality, which gives:

$$
(\mathbb{E}[\|\nabla L(w_{t-1,\hat{t}}; \mathcal{D})\|])^2 \leq \mathbb{E}[\|\nabla L(w_{t-1,\hat{t}}; \mathcal{D})\|^2].
$$

Substituting back into Eq.(29), we get:

$$
\Delta_{t+1} \leq (1 - \frac{\mu^2 \eta}{L})\Delta_t + \frac{L\eta\left(L\sigma_d^2 + \eta G^2\right)}{2}.
$$

The convergence becomes:

$$
\begin{aligned}
\Delta_{T_g} &\leq (1 - \frac{\mu^2 \eta}{L})^{T_g}\Delta_0 + \frac{L\eta\left(L\sigma_d^2 + \eta G^2\right)}{2}\sum_{t=0}^{T_g-1}(1 - \frac{\mu^2 \eta}{L})^t \\
&\leq \frac{\Delta_0}{1 + \frac{\mu^2 \eta}{L}T_g} + \frac{L^2\left(L\sigma_d^2 + \eta G^2\right)\left(1 - (1 - \mu\eta)^{T_g}\right)}{2\mu^2} \\
&\leq \frac{L\Delta_0}{\mu^2 \eta} \cdot \frac{1}{T_g} + \frac{L^2\left(L\sigma_d^2 + \eta G^2\right)}{2\mu^2}.
\end{aligned}
$$

Therefore, we conclude that:

$$\mathbb{E}\left[L(w_t; \mathcal{D})\right] - L(w_d^*; \mathcal{D}) \le \frac{C_t}{T_g} + B. \tag{30}$$

Thus, the remark of the convergence per communication round is proved. Then, we focus on the entire training process, which means the convergence of multi-round communications.

Let:
$$\mathbb{E}\left[L(w_t; \mathcal{D})\right] \le \alpha_t L(w_d^*; \mathcal{D}) + \beta_t \mathbb{E}\left[L(w_0; \mathcal{D})\right] + \gamma_t$$

Substituting into Eq. 30, we get:

$$\mathbb{E}\left[L(w_{t+1}; \mathcal{D})\right] \le L(w_d^*; \mathcal{D}) + \frac{L\mathbb{E}\left[L(w_t; \mathcal{D})\right]}{\mu^2 \eta} \cdot \frac{1}{T_g} + \frac{L^2\left(L\sigma_d^2 + \eta G^2\right)}{2\mu^2}$$

$$= (1 + \alpha_t \frac{L}{T_g \mu^2 \eta}) \cdot L(w_d^*; \mathcal{D}) + \frac{L}{T_g \mu^2 \eta} \beta_t \cdot \mathbb{E}\left[L(w_0; \mathcal{D})\right] + \left(\frac{L}{T_g \mu^2 \eta} \gamma_t + \frac{L^2\left(L\sigma_d^2 + \eta G^2\right)}{2\mu^2}\right) \tag{31}$$

Through Eq. 31, we get:
$$\begin{cases} \alpha_{t+1} = (1 + \alpha_t \frac{L}{T_g \mu^2 \eta}) \\ \beta_{t+1} = \frac{L}{T_g \mu^2 \eta} \beta_t \\ \gamma_{t+1} = \frac{L}{T_g \mu^2 \eta} \gamma_t + \frac{L^2\left(L\sigma_d^2 + \eta G^2\right)}{2\mu^2} \end{cases}$$

If the setting of $T_g$ is big enough, which means $L \ll T_g \mu^2 \eta \Rightarrow \frac{L}{T_g \mu^2 \eta} \ll 1$, with the initialization as $\alpha_0 = 0, \beta_0 = 1, \sigma_0 = 0$ we get:

$$\begin{cases} \alpha_t = \sum_{i=0}^{t-1}\left(\frac{L}{T_g \mu^2 \eta}\right)^i = \frac{T_g \mu^2 \eta}{T_g \mu^2 \eta - L}\left(1 - \left(\frac{L}{T_g \mu^2 \eta}\right)^t\right) \le \frac{T_g \mu^2 \eta}{T_g \mu^2 \eta - L} \approx 1 \\ \beta_t = \left(\frac{L}{T_g \mu^2 \eta}\right)^t \approx 0 \\ \gamma_t = \frac{L^2\left(L\sigma_d^2 + \eta G^2\right)}{2\mu^2} \cdot \frac{T_g \mu^2 \eta}{T_g \mu^2 \eta - L}\left(1 - \left(\frac{L}{T_g \mu^2 \eta}\right)^t\right) \le \frac{T_g \eta L^2\left(L\sigma_d^2 + \eta G^2\right)}{2(T_g \mu^2 \eta - L)} \end{cases}$$

Then, the entire convergence can be proved as:

$$E\left[L(w_t; \mathcal{D})\right] \le L(w_d^*; \mathcal{D}) + \frac{T_g \eta L^2(L\sigma_d^2 + \eta G^2)}{2(T_g \mu^2 \eta - L)}.$$

$\square$

## C ADDITIONAL EXPERIMENTAL RESULTS

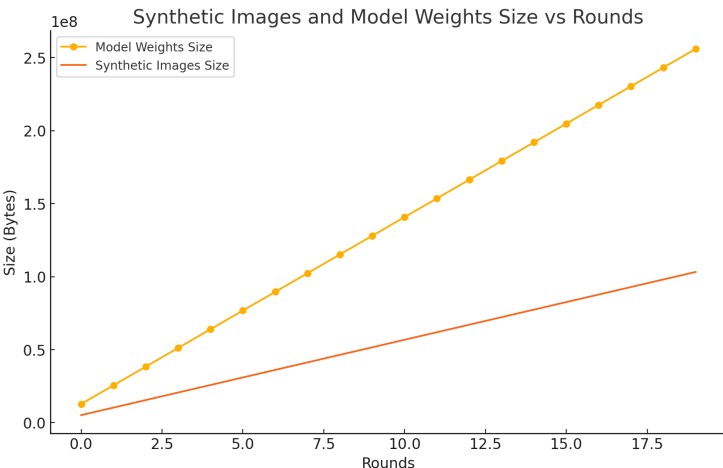

Figure 6: Comparison between synthetic images and model weights

Fig 6 shows how the size of the synthetic data and that of the model weight change with rounds. We can see that the communication efficiency improves while transferring the synthetic data (images) which is 2.5 times smaller than model weights.

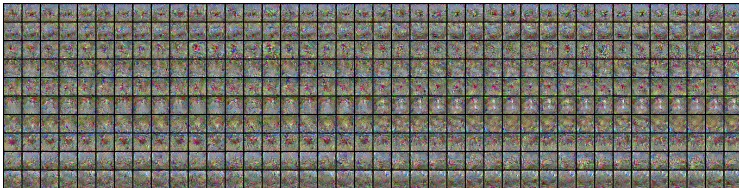

Figure 7: Synthetic images by our method

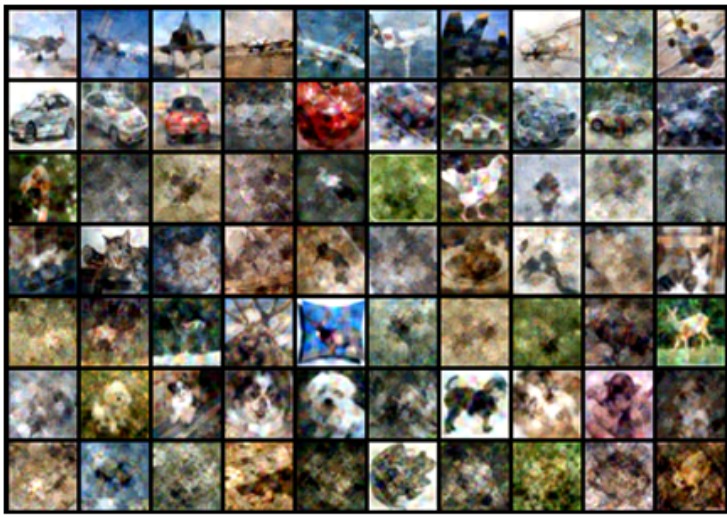

Figure 8: synthetic images by FedDM

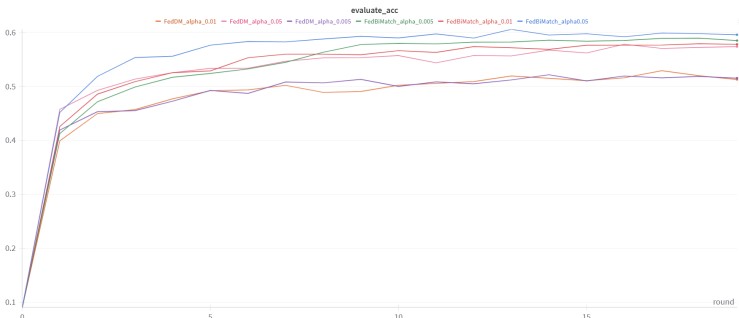

Figure 9: Comparison using real data

In FedDM, real client data is randomly selected as the initialization for synthetic images. However, privacy leakage is very likely to happen with this initialization. For example, a lot of image feature information can be observed in Fig 8. Thus, in our experiments, we opted to generate synthetic images from random initialization to prevent potential privacy leakage (see Fig 7). We also conducted additional experiments with real data initialization, where our method still outperforms FedDM (see Fig 9).

| Client Sampling | FedDM | FedBiMatch |
|---|---|---|
| 1.0 | 0.3784 | 0.4611 |
| 0.8 | 0.3542 | 0.4517 |
| 0.5 | 0.3525 | 0.4268 |
| 0.2 | 0.3584 | 0.4101 |

Table 3: Performance comparison between FedDM and FedBiMatch across client sampling rates.

Tab 3 shows that a lower participating rate will have lower accuracy. The setting is 50 clients with different join ratios.

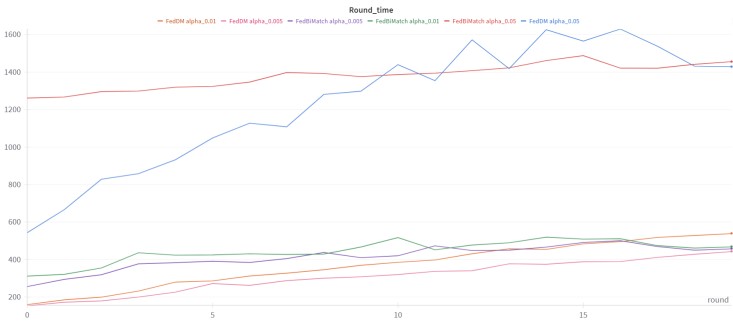

Figure 10: Computation overhead

Fig 10 shows that our computation on the server side(with gradient matching) doesn't cost a lot of computation overhead compared to FedDM. This is because the gradient matching only operates on synthetic data. The size of synthetic images depends on IPC. In our setting, each class only has 10 synthetic images. So the computation cost is still acceptable.

Fig 11 shows the performance on the larger number of devices. This is because, with fewer clients, the aggregated updates can be overly influenced by a small number of clients, especially in non-IID settings. Increasing the number of clients reduces the variance in updates and ensures a fairer representation of all client datasets

Tab C shows the class distribution for different $\alpha$'s in the Dirichlet distribution.

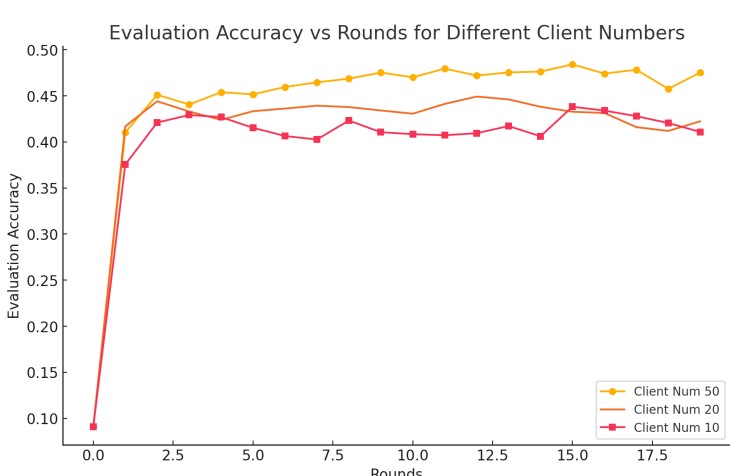

Figure 11: Comparison on large devices [the performance is not correct]

| Client | alpha=0.005 | alpha=0.01 | alpha=0.05 | alpha=0.1 |
|--------|-------------|------------|------------|-----------|
| client0 | [9] | [1, 8] | [1, 5, 9] | [1, 6, 7, 8] |
| client1 | [3] | [0, 5] | [0, 2, 3] | [0, 1, 2, 3] |
| client2 | [2] | [6] | [2, 7] | [1, 2] |
| client3 | [5] | [1, 7] | [6, 7, 8, 9] | [2, 7, 8, 9] |
| client4 | [8] | [3] | [0, 1, 3, 4] | [0, 1, 3, 4] |
| client5 | [0] | [2, 9] | [0, 6] | [0, 5, 6, 7, 8, 9] |
| client6 | [6] | [7] | [3, 5, 6] | [2, 3, 6] |
| client7 | [1] | [2, 4] | [1, 6, 8, 9] | [1, 2, 3, 5, 6, 7] |
| client8 | [4] | [0] | [0, 4, 5] | [0, 4, 6, 7, 8] |
| client9 | [7] | [0] | [1, 8] | [1, 5, 6, 8] |

Table 4: Client Classes for alpha=0.005, alpha=0.01, alpha=0.05 and alpha=0.1

