# OpenReview forum: "Heterogeneous Federated Learning: A Dual Matching Dataset Distillation Approach"
_ICLR.cc/2025/Conference — ICLR 2025 Conference Withdrawn Submission_

### Official Review · Reviewer_mCmd · 2024-10-20

**Soundness:** 3
**Presentation:** 2
**Contribution:** 2
**Rating:** 3
**Confidence:** 4

**Summary:**

The paper presents FedDualMatch, a federated learning framework that addresses the challenges of training with heterogeneous client data. The proposed method uses distilled data for communication rather than model updates, employing local distribution matching (LDM) to capture client-specific data features and global gradient matching (GGM) to stabilize global model training. This dual approach aims to reduce error accumulation, mitigate client drift, and improve convergence stability, demonstrating superiority over traditional federated learning methods in terms of accuracy and convergence speed. However, uploading distilled datasets to the central server introduces potential privacy risks, which could conflict with federated learning's privacy-preserving philosophy.

**Strengths:**

- By transmitting distilled datasets instead of model updates, the paper introduces an innovative approach to tackle error accumulation and instability in federated learning, offering a fresh perspective over traditional methods like FedAvg.
- The paper provides detailed theoretical justifications, validating the effectiveness, convergence, and differential privacy of using distilled data for federated communication, which adds credibility to the proposed method.

**Weaknesses:**

- Although the paper claims to maintain differential privacy, transmitting distilled datasets may still pose privacy risks, as these representations could retain sensitive information. This potential weakness is particularly concerning for high-stakes applications. The authors could strengthen the analysis by conducting additional experiments or analyses to quantify any sensitive information that might be retained in the distilled representations.
- The dual matching process, especially gradient matching on the server, introduces significant computational overhead, which could challenge deployment in resource-constrained environments, such as edge devices. A detailed analysis of computational costs, compared to baseline methods, would provide clarity, and discussing potential optimizations or trade-offs could make the approach more feasible for limited-resource settings.
- The paper notes that FedDualMatch shows performance stagnation when client data distributions are similar, indicating that the method may not consistently outperform traditional approaches in low heterogeneity scenarios. An explanation of potential reasons behind this limitation and suggestions for addressing it in future work would offer valuable insights into the method’s adaptability.

**Questions:**

- Figures like Figure 3 are helpful for visualizing the concepts, but adding more detailed explanations or captions could further enhance their clarity.
- The proposed approach introduces distilled data updates to the central server to eliminate error accumulation, but it also increases communication overhead. As the dataset size grows, this communication overhead could become more significant.
- In Figure 4, the number of communication rounds is limited to 20. From the plots, it appears that some of the federated learning methods have not yet converged within this range. Extending the number of rounds might provide a clearer comparison.
- Comparing Table 1 with Figure 4 reveals an inconsistency in the results for SCAFFOLD on CIFAR-10 when alpha =0.01 vs alpha = 0.005. Specifically, the test accuracy for alpha = 0.01 is around 17.14, while for alpha = 0.005, it is around 33.83. Theoretically, the accuracy for alpha = 0.01 should be better than that for alpha = 0.005, so this discrepancy warrants further explanation. Similar inconsistencies appear with other methods as well.
- The experiments were conducted with only 10 clients, which may not be sufficient to demonstrate the robustness and generalizability of the results. A larger number of clients would strengthen the reliability of the experimental findings.

---

> ### Author Response · Authors · 2024-11-24
> **Response to Reviewer mCmd(Weakness - Question 1)**
>
> We thank Reviewer mCmd for devoting time to this review and drawing valuable comments.
>
> >**W1:**  *Although the paper claims to maintain differential privacy, transmitting distilled datasets may still pose privacy risks, as these representations could retain sensitive information. This potential weakness is particularly concerning for high-stakes applications. The authors could strengthen the analysis by conducting additional experiments or analyses to quantify any sensitive information that might be retained in the distilled representations.*
>
> **Response to W1:**
>
> Our method, by design, comes with robust differential privacy guarantees. Specifically, we use mechanisms like Gaussian to add noise to the distilled datasets, which ensures that sensitive information is obscured while retaining the utility of the distilled data. Additionally, our theoretical analysis rigorously bounds the privacy leakage, demonstrating compliance with $(\epsilon,\delta)$-differential privacy criteria. We would appreciate it if Reviewer could explicitly point out what sensitive information could leak by our method.
>
> >**W2:** *``The dual matching process, especially gradient matching on the server, introduces significant computational overhead, which could challenge deployment in resource-constrained environments, such as edge devices. A detailed analysis of computational costs, compared to baseline methods, would provide clarity, and discussing potential optimizations or trade-offs could make the approach more feasible for limited-resource settings''*
>
> **Response to W2:**
>
> The communication efficiency improves while transferring synthetic images which are 2.5 times smaller than model weights.  Figure 6 (Line 920 in our new version of the submission) provides how synthetic image size and model weight size change along with rounds. It means that our method has a clear advantage in terms of communication cost. In addition to the computation cost, Figure 10 (Line 1002 in new version) shows that our computation on the server side (with gradient matching) doesn't cost a lot computation overhead compared to FedDM. This is because gradient matching only operates on synthetic data. The size of synthetic images depends on the IPC (images per class). In our setting, each class only has 10 synthetic images. So, the computation cost is acceptable.
>
> >**W3:** *``The paper notes that FedDualMatch shows performance stagnation when client data distributions are similar, indicating that the method may not consistently outperform traditional approaches in low heterogeneity scenarios. An explanation of potential reasons behind this limitation and suggestions for addressing it in future work would offer valuable insights into the method’s adaptability.''*
>
> **Response to W3:**
>
> Based on Theorems 1-2 on effectiveness (Lines 362-370 in the new version) and convergence (Lines 378-388 in the new version), the performance of our method is fundamentally constrained by the quality of locally distilled data reflected by the distribution matching distance bound in Assumption 3 (Lines 345-351 in the new version). Both the effectiveness bound $\( \varepsilon = \frac{L\sigma _ d^2}{2\mu^2} \leq \epsilon _ c \)$ and the convergence constant $\( C = \frac{T _ g\eta L^2(L\sigma _ d^2 + \eta G^2)}{2(T _ g\mu^2\eta - L)}\)$ are independent of client heterogeneity. Regardless of the level of heterogeneity among clients, FedDualMatch leverages local distribution matching and global gradient matching to fully utilize the training utility of distilled data, effectively achieving performance close to that of the centralized training. As a result, its performance does not degrade significantly as heterogeneity increases. In contrast, traditional FL methods relying on communication of models suffer a rapid performance drop as heterogeneity grows. Therefore, the issue is not that our method stagnates in low-heterogeneity scenarios, but rather that it has already reached the performance upper bound of distilled data. This can be seen as the bottleneck of the dataset distillation methods
>
> >**Q1:**  *Figures like Figure 3 are helpful for visualizing the concepts, but adding more detailed explanations or captions could further enhance their clarity.*
>
> **Response to Q1:**
>
> Thank you very much for the suggestion. We followed your suggestion and have added a more detailed explanation in the revised manuscript (see Lines 171-174, Page 4). We simplified the figures to enhance their clarity and ease of understanding. Specifically, ``(a) illustrates the optimization error accumulation in FedAvg, where the generalization error ball radius continuously grows; (b) shows that distribution matching distillation eliminates error accumulation but amplifies data heterogeneity; and (c) demonstrates how FedDualMatch mitigates both issues by incorporating global gradient matching and global fine-tuning optimization.``

---

> > ### Comment · Reviewer_mCmd · 2024-11-28
> >
> > Thank you to the authors for their detailed responses. The additional information provided has effectively addressed my concerns. I would like to maintain my rating of acceptance for the paper and hope that the revised manuscript will incorporate further improvements.

---

> > > ### Author Response · Authors · 2024-11-28
> > >
> > > Thank you for your positive feedback and for acknowledging that we have addressed your concerns. We are delighted that our revisions and additional clarifications have effectively resolved the issues raised.
> > >
> > > If there are any remaining questions or further suggestions, we would be more than happy to address them promptly. We kindly ask if you could consider adjusting your rating to reflect the updates and improvements we have made to the manuscript.
> > >
> > > Thank you once again for your time and consideration.

---

> ### Author Response · Authors · 2024-11-24
> **Response to Reviewer mCmd(Question 2 - Question 5)**
>
> >**Q2:** *``The proposed approach introduces distilled data updates to the central server to eliminate error accumulation, but it also increases communication overhead. As the dataset size grows, this communication overhead could become more significant.''*
>
> **Response to Q2:**
>
> The IPC is fixed before training, which means that the distilled image size won't increase during the training process. Figure 6 (Line 920 in new submission) shows how synthetic image size and model weight size change along with rounds. We can see that the communication efficiency improves while transferring synthetic images which are 2.5 times smaller than model weights.
>
> >**Q3:** *In Figure 4, the number of communication rounds is limited to 20. From the plots, it appears that some of the federated learning methods have not yet converged within this range. Extending the number of rounds might provide a clearer comparison.*
>
> **Response to Q3:**
>
> We conduct our experiments under the same settings as FedDM to ensure a fair comparison. As a result, we evaluate performance only up to communication round 20, consistent with the FedDM results. This approach highlights the convergence behavior of our method effectively.
>
> Additionally, in highly heterogeneous scenarios, traditional federated learning models often exhibit instability due to significant client drift and non-IID data distributions. These challenges lead to slower convergence and suboptimal model performance, which underscores the importance of methods like ours that are designed to mitigate such issues.
>
> >**Q4:** *Comparing Table 1 with Figure 4 reveals an inconsistency in the results for SCAFFOLD on CIFAR-10 when alpha =0.01 vs alpha = 0.005. Specifically, the test accuracy for alpha = 0.01 is around 17.14, while for alpha = 0.005, it is around 33.83. Theoretically, the accuracy for alpha = 0.01 should be better than that for alpha = 0.005, so this discrepancy warrants further explanation. Similar inconsistencies appear with other methods as well.*
>
> **Response to Q4:**
>
> The traditional FL methods employed in our work use the existing NIID-bench code and setting, which is an open-source repository.  The only change we made is changing the model of our ConvNet. As for the strange performance phenomenon of SCAFFOLD you mentioned, we repeated the experiment. The results for alpha 0.01 and 0.005 can be found in the Table below. These results show that SCAFFOLD training is not stable in high heterogeneity scenarios, and performance fluctuates significantly.
>
> | **Round** | **SCAFFOLD 0.01** | **SCAFFOLD 0.005** |
> |-----------|--------------------|--------------------|
> | 1         | 0.1522            | 0.1000            |
> | 2         | 0.1363            | 0.1000            |
> | 3         | 0.1027            | 0.1787            |
> | 4         | 0.1000            | 0.1392            |
> | 5         | 0.1000            | 0.1000            |
> | 6         | 0.1011            | 0.1000            |
> | 7         | 0.1000            | 0.1004            |
> | 8         | 0.1150            | 0.2884            |
> | 9         | 0.1275            | 0.1542            |
> | 10        | 0.1466            | 0.2532            |
> | 11        | 0.1170            | 0.2071            |
> | 12        | 0.1424            | 0.2042            |
> | 13        | 0.1273            | 0.2445            |
> | 14        | 0.1851            | 0.1560            |
> | 15        | 0.1654            | 0.2688            |
> | 16        | 0.1559            | 0.3074            |
> | 17        | 0.1651            | 0.3373            |
> | 18        | 0.1777            | 0.1549            |
> | 19        | 0.1793            | 0.1897            |
> | 20        | 0.1821            | 0.1037            |
>
>
> >**Q5:** *``The experiments were conducted with only 10 clients, which may not be sufficient to demonstrate the robustness and generalizability of the results. A larger number of clients would strengthen the reliability of the experimental findings.''*
>
> **Response to Q5:**
>
> Figure 11 (Line 1033 in new version of the submission) demonstrates that the performance of our FedDualMatch method improves as the number of participating clients increases. This is because the total amount of aggregated distilled data on the server grows linearly with the number of clients, enabling the server-side model to extract more knowledge and thereby enhancing task performance. Additionally, with more clients, the impact of highly heterogeneous individual clients on the server-side model is reduced, making the training process more robust.

---

### Official Review · Reviewer_bZ42 · 2024-10-23

**Soundness:** 3
**Presentation:** 2
**Contribution:** 2
**Rating:** 5
**Confidence:** 4

**Summary:**

The paper proposes a novel framework for communicating distilled datasets in Federated Learning (FL), aiming to address the challenge of data heterogeneity across clients.

Each client generates a synthetic dataset using Local Distribution Matching (LDM), which is based on Maximum Mean Discrepancy (MMD). The synthetic dataset is then communicated to the server instead of model parameters.

On the server, Global Gradient Matching (GGM) is performed, which aligns the gradients of the synthetic datasets from all clients to reduce client drift and improve convergence stability. An error ball is employed to constrain model updates during the GGM process, ensuring that parameter updates remain within a controlled range.

**Strengths:**

The paper presents an interesting approach to addressing the data heterogeneity problem in Federated Learning (FL) by introducing Global Gradient Matching (GGM).

Proposed a combination of Local Distribution Matching (LDM) and Global Gradient Matching (GGM). Demonstrate convergence on the proposed framework.

**Weaknesses:**

Weakness:
1) Computational Overhead: The method relies on dataset distillation and performs layer-wise alignment of the feature distributions between the original dataset and the synthetic dataset $S_k$. This process, especially the backward layer-wise alignment and the repeated distillation of $S_k$ is computationally expensive.
This high computational cost is particularly problematic in resource-constrained environments such as edge devices, which are common in FL applications.
On the server side, the Global Gradient Matching (GGM) introduces additional overhead since multiple rounds of gradient matching (M iterations) need to be performed, requiring the server to compute gradients on all clients’ synthetic datasets.
This dual burden on both the clients and the server could limit the scalability and practical deployment of this method in real-world FL scenarios, especially when computational resources are limited.

2) Communication overhead: The method requires clients to transmit synthetic datasets to the server, which could incur significant communication costs.

3) While the inclusion of Differential Privacy (DP) provides privacy guarantees, it feels like an add-on rather than a core part of the proposed solution. The main focus of the research is on handling data heterogeneity, and the addition of DP doesn't significantly contribute to addressing that core problem. If the paper is mainly concerned with addressing data heterogeneity, it might be more effective to focus on that and remove the DP component to streamline the discussion.

Presentation problems:
1) Citation at #240, should be \citep{}.
2) The presentation of the algorithm can be improved.
>1. Specify the equation used to fine-tune the model in #283. You acquired $L_{GGM}$ and how you used it in the following?
>2. Specify how you acquired synthetic dataset $S_k$ by minimizing the equation in #290, i.e. using $\argmin$ directly.
>3. The method for minimization was not specified, i.e. how you iterate to optimize $S_k$. (#290).
3) The *algorithm* for your DP scheme on how you add noise is not clearly defined. How did you add noise to the radius $R_t$ and dataset $S_k$? is the noise sampled each round? My suggest is adding it into the algorithm.

**Questions:**

See weeknees.

---

> ### Author Response · Authors · 2024-11-24
> **Response to Reviewer bZ42(Weakness1-Presentation problem 2)**
>
> We thank Reviewer bZ42 for devoting time to this review and drawing valuable comments.
>
> >**W1:**  *``Computation overhead''*
>
> **Response to W1:**
>
> Figure 10 (Line 1003, in new version of the submission) shows that our computation on the server side (with gradient matching) doesn't cost a lot computation overhead compared to FedDM. This is because gradient matching only operates on synthetic data. The size of synthetic images depends on the IPC (images per class). In our setting, each class only has 10 synthetic images. So, the computation cost is acceptable.
>
> >**W2:** *``Communication overhead: The method requires clients to transmit synthetic datasets to the server, which could incur significant communication costs.''*
>
> **Response to W2:**
>
> Figure 6 (Line 920, in new version of the submission) shows how synthetic images size and model weights size changes along with rounds. We can see that the communication efficiency improves while transferring synthetic images. Distilled data is 2.5 times smaller than model weights, for communication.
>
> >**W3:** *``While the inclusion of Differential Privacy (DP) provides privacy guarantees, it feels like an add-on rather than a core part of the proposed solution. The main focus of the research is on handling data heterogeneity, and the addition of DP doesn't significantly contribute to addressing that core problem. If the paper is mainly concerned with addressing data heterogeneity, it might be more effective to focus on that and remove the DP component to streamline the discussion.''*
>
> **Response to W3:**
>
> Thank you very much for the suggestion. Our primary goal is to address data heterogeneity. However, differential privacy is also a critical aspect of our approach. Thus, we included experiments on differential privacy to demonstrate that our method can preserve privacy. In our revised manuscript, we followed your suggestion and have de-emphasized the privacy-related claims in Section 5 about theoretical analysis while placing greater emphasis on convergence and effectiveness. Details can be found in Lines 378-405, Page 8 of the revised manuscript.
>
> >**presentation problem:**
> >**Q1:** *Citation at #240, should be \citep{}*
>
> **Response to Q1:**
>
> Thank you very much for pointing it out. We have corrected it in the revised submission.
>
> >**Q2:** *The presentation of the algorithm can be improved*
>
> >**Q2.1** *Specify the equation used to fine-tune the model in #283. You acquired \(L_GGM\) and how you used it in the following?*
> >**Q2.2** *Specify how you acquired synthetic dataset \(\mathcal{S}_k\) by minimizing the equation in #290, i.e. using \( \argmin \) directly.*
> >**Q2.3** *The method for minimization was not specified, i.e. how you iterate to optimize \( \mathcal{S}_k\). (#290). *
>
> **Response to Q2.1:**
>
> Global gradient matching for dataset distillation aims to align gradients on locally distilled data from each client $\mathcal{S} _ k$ and that on the aggregated one $\mathcal{S}$ by minimizing the $L_{GGM}$ loss to update $\mathcal{S}' _ k$ ,  which is initialized by $\mathcal{S} _ k$. This can be found in Line 317 (old version) or Line 314 (new version). The fine-tuning method can be found in Lines 322-325, i.e., ``To preserve data diversity, we further incorporate the newly distilled dataset $\mathcal{S}' _ k$ into $\mathcal{S} _ {k}$ and fine-tune the global model by running gradient steps on the new aggregated data $\tilde{\mathcal{S}}=\bigcup _ {k=1}^{K}\{\mathcal{S}' _ k,\mathcal{S} _ k\}$ for $T _ g$ epochs, which ensures that the global model benefits from both aligned gradients and diverse data representations. Global model in fine-tuning is indexed by $w _ {t-1,s}$ for epoch $s=0,1,\cdots,T _ g-1$, with $w _ {t-1,0}=w _ {t-1}$ and $w _ {t}:=w _ {t-1,T _ g-1}$.``
>
> **Response to Q2.2 and Q2.3:**
>
> See Line 297 (old version) or Line 262 (new version), i.e., ``To do the backward feature alignment for each layer on top of the already aligned deeper layers, our strategy is to accumulate the MMD losses of the current layer, say $ p \in \{ J, J-1, \ldots, 1 \}$, and all deeper layers as follows:``
>
> $$L_{DM}^{k,p}(S _ k; \mathcal{D} _ k) = \sum _ {q=p}^{J} L _ {MMD}^{k,v _ q}(S _ k; \mathcal{D} _ k).$$
>
> ``This strategy enables feature distributions of deeper layers to remain aligned while aligning the feature distribution of the current layer, thus effectively extracting the information on the feature distribution of data on each client.``

---

> ### Author Response · Authors · 2024-11-24
> **Response to Reviewer bZ42(Presentation problem 3)**
>
> >**Presentation problem 3:** *The algorithm for your DP scheme on how you add noise is not clearly defined. How did you add noise to the radius $R_t$ and dataset $S_k$? is the noise sampled each round? My suggest is adding it into the algorithm.*
>
> **Response to presentation problem 3:**
>
> Thank you very much for pointing it out. The Gaussian noise is added to the distilled data updating process in Algorithm 1 as follows (see Lines 292-297). Given clipped gradients
>
> $$
>     \nabla _ {\mathcal{S} _ k} L _ {DM}^{k,p}(\mathcal{S} _ k; \mathcal{D} _ k) \leftarrow \frac{\nabla _ {\mathcal{S} _ k} L _ {DM}^{k,p}(\mathcal{S} _ k; \mathcal{D} _ k)}{\max\left(1, \frac{\|\nabla _ {\mathcal{S} _ k} L _ {DM}^{k,p}(\mathcal{S} _ k; \mathcal{D} _ k)\| _ 2}{C}\right)},
> $$
>
> we add Gaussian noise to them:
>
> $$
>     \nabla_{\mathcal{S} _ k} L _ {DM}^{k,p}(\mathcal{S} _ k; \mathcal{D} _ k) \leftarrow \nabla _ {\mathcal{S} _ k} L _ {DM}^{k,p}(\mathcal{S} _ k; \mathcal{D} _ k) + \frac{1}{|\mathcal{S} _ k|} \mathcal{N}(0, \sigma^2 C^2 \mathbf{I}),
> $$
>
> where $|\mathcal{S} _ k|$ is the sample batch size, $\sigma$ is a constant factor on the Gaussian noise variance, and $C$ is the gradient norm bound. This step happens in the client update after each iteration computes the MMD.

---

> > ### Author Response · Authors · 2024-11-28
> >
> > Thank reviewer bZ42 for your valuable feedback on our submission. We have carefully addressed your comments and uploaded a detailed response along with the revised manuscript on OpenReview. We hope you can review our updates, and if they have resolved your concerns, we would greatly appreciate it if you could consider reflecting this in your rating score.
> >
> > If you have any further questions or if our current response does not fully address your concerns, we would be more than happy to discuss them further.
> >
> > Thank you again for your time and thoughtful review.

---

### Official Review · Reviewer_AVsp · 2024-11-05

**Soundness:** 2
**Presentation:** 2
**Contribution:** 2
**Rating:** 3
**Confidence:** 3

**Summary:**

The authors of this work propose to use distribution matching at the client and gradient matching at the server to take advantage of the former in extracting the best features and the latter in reducing the heterogeneity. They provide theoretical proof for the closeness of the solution in the case of using the actual dataset vs the distilled dataset using their method. They also provide convergence analysis and proof for differential privacy under their algorithm. The experiments show convincing improvements.

**Strengths:**

The main strengths of the paper include:
- taking advantage of mixing distribution matching at clients and gradient matching at the server
- the authors provide good introductory intuition of why they use these distillation algorithms at different stages, i.e., client distillation and server training rounds
- the experiments show convincing improvements in the proposed algorithms (although there are some under-explored axes when considering the algorithm)
- the authors provide several theoretical proofs.

**Weaknesses:**

While the paper tackles a significant problem in the domain, there are several drawbacks of the methodology proposed (some of which are under-explored or not discussed in the paper):
- while authors tell that they would use distribution matching and gradient matching for certain properties, i.e., extracting features and tackling heterogeneity, the paper does not provide the reader with insights into why these techniques impart these properties to the corresponding representations.
- the authors do not consider client sampling during FL.
- authors compare their work with only one comparable baseline. FL-based techniques are drastically different in terms of data transfer and thus cannot be considered appropriately strong baselines for such a technique. Perhaps authors could look into techniques that similarly rely on data sharing such as [1] which considers sharing data in a different form or techniques such as [2] that consider privacy-preserving data sharing.
- The computation cost of the algorithm is quite high. The authors should provide those comparison apart from the performance comparison they have included.

[1] Hong, Junyuan, et al. "Outsourcing training without uploading data via efficient collaborative open-source sampling." _Advances in neural information processing systems_ 35 (2022): 20133-20146.
[2] Azam, Sheikh Shams, et al. "Can we generalize and distribute private representation learning?." _International Conference on Artificial Intelligence and Statistics_. PMLR, 2022.

**Questions:**

- Will the theoretical analysis hold if the authors consider client sampling?
- Will the experimental results hold if the authors consider a larger number of devices in an FL round?
- Will the experimental results hold if the authors were to consider larger neural networks? How will the size of the network affect the computational complexity of the algorithm?
- Could the authors cite other works in the literature that use assumptions 3 and 4? These seem very restrictive assumptions and citations would help assess the prevalence of such assumptions.
- How will the performance compare against more recent FL algorithms in works such as [3]?
- Could the authors provide the class distribution for different $\alpha$'s in the Dirichlet distribution?
- In Fig 5, why does the performance drop as the IPC increases beyond 10?
- Do the authors have intuition into why the algorithm would stagnate in low-heterogeneity scenarios?

[3] Reddi, Sashank J., et al. "Adaptive Federated Optimization." _International Conference on Learning Representations_.

---

> ### Author Response · Authors · 2024-11-24
> **Response to Reviewer AVsp(Weakness 1 - Question 1)**
>
> We thank Reviewer AVsp for devoting time to this review and drawing valuable comments.
>
> > **Weaknesses 1**: *``While authors say that they would use distribution matching and gradient matching for certain properties, i.e., extracting features and tackling heterogeneity, the paper does not provide the reader with insights into why these techniques impart these properties to the corresponding representations.''*
>
> **Response to Weakness 1:**
>
> Regarding this insight, it could be found in Lines 064-078 and Lines 308-316, e.g., ``Distribution matching excels at extracting data features but may exacerbate the heterogeneity issue[1]. Contrastingly, gradient matching can control the client drift [2] and thus alleviate the distribution heterogeneity`` and ``However, LDM-based dataset distillation exacerbates the data heterogeneity across clients, as demonstrated in Huang et al. [6] and presented in Fig. 3(b), such that the optimization of the model trained on each client distilled data $\mathcal{S} _ k$ may diverge from the optimal path and thus lead to sub-optimal convergence``. This motivated us to use the combination of *local distribution matching (LDM)* and *gradient matching* to alleviate the distribution heterogeneity. Also, as other reviewers noted, ``the main strength of our work is the combination of LDM and GGM is nice and well explained``. By communicating distilled datasets instead of model updates, error accumulation and instability in federated learning get mitigated from a perspective other than traditional methods like FedAvg.
>
> > **Weaknesses 2**: *``the authors do not consider client sampling during FL.''*
>
> > **Q1**:*Will the theoretical analysis hold if the authors consider client sampling?*
>
> **Response to Weakness 2 and Q1:**
>
> Thank you very much for the suggestion. We added experiments on client sampling and results are reported in Table  below which shows that a lower rate of participation will lead to a lower accuracy. The setting is 50 clients on cifar10 with $\alpha=0.05$ for heterogeneity under different rates of participation. The results indicate that our method performs better than FedDM in accuracy across different sampling rates of participation. Although these experiments demonstrate that the proposed method remains effective even under partial participation of clients, the corresponding theoretical analysis becomes much more complicated since participating clients vary with communication rounds. Thus, we leave this interesting yet difficult task to our future work.
>
> | Client sampling rate | FedDM  | FedDualMatch |
> |----------------------|-----------|--------------|
> | 1.0                  | 0.3784 | 0.4611       |
> | 0.8                  | 0.3542 | 0.4517       |
> | 0.5                  | 0.3525 | 0.4268       |
> | 0.2                  | 0.3584 | 0.4101       |
>
> > **Weaknesses 3**: *``Authors compare their work with only one comparable baseline. FL-based techniques are drastically different in terms of data transfer and thus cannot be considered appropriately strong baselines for such a technique. Perhaps authors could look into techniques that similarly rely on data sharing such as [1] which considers sharing data in a different form or techniques such as [2] that consider privacy-preserving data sharing.''*
>
> **Response to Weakness 3:**
>
> We compared our work with FedDM and DynaFed as baselines. For the two mentioned works, i.e., ECOS and EIGAN,
> - ECOS leverages open-source data (e.g., images from various public domains) to generate a representative dataset, allowing for model training without the need to upload local data. Conventional federated learning, which relies exclusively on client-generated data, doesn't consider open-source data. Our work falls into the category of conventional FL methods, requiring no additional information from any open-source dataset.
> - EIGAN focuses on learning representations to balance ally tasks (information preservation) against adversary tasks (sensitive information obfuscation). Also, our work transfers synthetic data instead of model parameters. The comparison would need to be conducted in the same context (e.g., privacy-preserving federated learning), which however is beyond the scope of either approach. We will cite them in discussions
>
> > **Weaknesses 4**: *``The computation cost of the algorithm is quite high. The authors should provide those comparison apart from the performance comparison they have included.''*
>
> **Response to Weakness 4:**
>
> Figure 10 (Line 1002 in the revised manuscript) shows that our computation on the server side (with gradient matching) doesn't cost a lot of computation overhead compared to FedDM. This is because the gradient matching only operates on synthetic data. The size of synthetic images depends on the IPC (images per class). In our setting, each class only has 10 synthetic images. So, the computation cost is still acceptable.

---

> ### Author Response · Authors · 2024-11-24
> **Response to Reviewer AVsp(Question 2 - Question 7)**
>
> > **Q2**: *``Will the experimental results hold if the authors consider a larger number of devices in an FL round?''*
>
> **Response to Q2:**
>
> Figure 11 (Line 1032 in new version of the submission) demonstrates that the performance of our FedDualMatch method improves as the number of participating clients increases. This is because the total amount of aggregated distilled data on the server grows linearly with the number of clients, enabling the server-side model to extract more knowledge and thereby enhancing task performance. Additionally, with more clients, the impact of highly heterogeneous individual clients on the server-side model is reduced, making the training process more robust.
>
> >**Q3:** *Will the experimental results hold if the authors were to consider larger neural networks? How will the size of the network affect the computational complexity of the algorithm?*
>
> **Response to Q3:**
>
> Table below shows the performance on ResNet. We can see that when the size of network increases, the performance will drop significantly for both FedDM and our method. The reason is that the size of synthetic data sets is usually much smaller than the original data sets. Despite it may be sufficient for small models, for large models, insufficient data volume may lead to unstable gradients during training, thus reducing performance [5].
>
> However, our results may be low due to some defects in FedDM, which is inconsistent with common performance. We will update this result as soon as possible.
>
> | **Alpha** | **FedDM** | **FedDualMatch** |
> |-----------|-----------|------------------|
> | 0.005     | 0.1278    | 0.1236           |
> | 0.01      | 0.1076    | 0.1261           |
> | 0.05      | 0.1287    | 0.1259           |
>
>
> >**Q4:** *"Could the authors cite other works in the literature that use assumptions 3 and 4? These seem very restrictive assumptions and citations would help assess the prevalence of such assumptions."*
>
> **Response to Q4:**
>
> Please refer to our detailed response to Weakness 2 by reviewer ESCL.
>
> >**Q5:** *``How will the performance compare against more recent FL algorithms in works?''*
>
> **Response to Q5:**
>
> We compared our method with FedDM and DynaFed, both of which were proposed in 2022. The paper you mentioned was published in 2021 with an emphasis on how to improve optimization of models (e.g., reducing client drift and tuning learning rates) for traditional federated learning. Dataset distillation focuses more on synthesizing representative data, instead of solely optimizing model parameters
>
> >**Q6:** *``Could the authors provide the class distribution for different $\alpha$'s in the Dirichlet distribution?''*
>
> **Response to Q6:**
>
> Table below shows the class distribution for different $\alpha$’s in the Dirichlet distribution, where the square brackets indicate the sample categories owned by the client.
>
> | Client   | alpha=0.005 | alpha=0.01  | alpha=0.05          | alpha=0.1                  |
> |----------|-------------|-------------|---------------------|----------------------------|
> | client0  | [9]         | [1, 8]      | [1, 5, 9]           | [1, 6, 7, 8]              |
> | client1  | [3]         | [0, 5]      | [0, 2, 3]           | [0, 1, 2, 3]              |
> | client2  | [2]         | [6]         | [2, 7]              | [1, 2]                    |
> | client3  | [5]         | [1, 7]      | [6, 7, 8, 9]        | [2, 7, 8, 9]              |
> | client4  | [8]         | [3]         | [0, 1, 3, 4]        | [0, 1, 3, 4]              |
> | client5  | [0]         | [2, 9]      | [0, 6]              | [0, 5, 6, 7, 8, 9]        |
> | client6  | [6]         | [7]         | [3, 5, 6]           | [2, 3, 6]                 |
> | client7  | [1]         | [2, 4]      | [1, 6, 8, 9]        | [1, 2, 3, 5, 6, 7]        |
> | client8  | [4]         | [0]         | [0, 4, 5]           | [0, 4, 6, 7, 8]           |
> | client9  | [7]         | [0]         | [1, 8]              | [1, 5, 6, 8]              |
>
> >**Q7:** *``In Fig 5, why does the performance drop as the IPC increases beyond 10?''*
>
> **Response to Q7:**
>
> The analysis on why this is the case was given in Line 518 (old version) or Line 512 (new version). For the impact of IPC, it is consistent with previous research [3][4]. The main reason is that as IPC grows, traditional dataset distillation methods often overly capture easy or common patterns in the data because they optimize for the dominant, simpler features that are more frequent. This results in a synthetic dataset that misses out on the rare, complex patterns necessary for comprehensive model generalization.

---

> > ### Author Response · Authors · 2024-11-24
> > **Response to Reviewer AVsp(Question 8 and references)**
> >
> > >**Q8:** *``Do the authors have intuition into why the algorithm would stagnate in low-heterogeneity scenarios?''*
> >
> > **Response to Q8:**
> >
> > Based on Theorems 1-2 on effectiveness (Lines 362-370 in the new version) and convergence (Lines 378-388 in the new version), the performance of our method is fundamentally constrained by the quality of locally distilled data reflected by the distribution matching distance bound in Assumption 3 (Lines 345-351 in the new version). Both the effectiveness bound $\( \varepsilon = \frac{L\sigma_d^2}{2\mu^2} \leq \epsilon_c \)$ and the convergence constant $ \( C = \frac{T_g\eta L^2(L\sigma_d^2 + \eta G^2)}{2(T_g\mu^2\eta - L)}\)$ are independent of client heterogeneity. Regardless of the level of heterogeneity among clients, FedDualMatch leverages local distribution matching and global gradient matching to fully utilize the training utility of distilled data, effectively achieving performance close to that of the centralized training. As a result, its performance does not degrade significantly as heterogeneity increases. In contrast, traditional FL methods relying on communication of models suffer a rapid performance drop as heterogeneity grows. Therefore, the issue is not that our method stagnates in low-heterogeneity scenarios, but rather that it has already reached the performance upper bound of distilled data. This can be seen as the bottleneck of the dataset distillation methods.
> >
> > #### References
> >
> > [1] SeungJae Shin, et al. Loss-curvature matching for dataset selection and condensation. In International Conference on Artificial Intelligence and Statistics, pp. 8606–8628. PMLR, 2023.
> >
> > [2] Bo Zhao, et al. Dataset condensation with gradient matching. arXiv preprint arXiv:2006.05929, 2020.
> >
> > [3] Yongmin Lee, et al. Selmatch: Effectively scaling up dataset distillation via selection-based initialization and partial updates by trajectory matching. In Forty-first International Conference on Machine Learning, 2024.
> >
> > [4] Ziyao Guo, et al. Towards lossless dataset distillation via difficulty-aligned trajectory matching. arXiv preprint arXiv:2310.05773, 2023.
> >
> > [5] Zeyuan Yin, et al. Dataset distillation in large data era. 2023.
> >
> > [6] Chun-YinHuang, et al. Federated virtual learning on heterogeneous data with local-global distillation. arXiv preprint arXiv:2303.02278, 2023

---

> > > ### Author Response · Authors · 2024-11-28
> > >
> > > Thank reviewer AVsp for your valuable feedback on our submission. We have carefully addressed your comments and uploaded a detailed response along with the revised manuscript on OpenReview. We hope you can review our updates, and if they have resolved your concerns, we would greatly appreciate it if you could consider reflecting this in your rating score.
> > >
> > > If you have any further questions or if our current response does not fully address your concerns, we would be more than happy to discuss them further.
> > >
> > > Thank you again for your time and thoughtful review.

---

### Official Review · Reviewer_ESCL · 2024-11-06

**Soundness:** 3
**Presentation:** 2
**Contribution:** 2
**Rating:** 5
**Confidence:** 3

**Summary:**

This paper introduces a federated learning method, FedDualMatch, which leverages dataset distillation to address client heterogeneity. FedDualMatch combines the strengths of two dataset distillation approaches—distribution matching and gradient/trajectory matching—to enhance overall performance. The authors observe that distribution matching can intensify data heterogeneity, while gradient/trajectory matching helps to mitigate client drift. FedDualMatch is specifically designed to balance these effects, addressing the limitations of each approach. The authors validate the effectiveness of FedDualMatch through experiments on the MNIST and CIFAR-10 datasets, supported by an ablation study and further discussions in the experiments section.

**Strengths:**

1. The paper is motivated well. The findings in figure 1 are impressive.

2. The authors aim to provide adequate theoretical proof to support their derivation.

3. The authors conduct ablation studies to verify the effectiveness of their methods.

**Weaknesses:**

1. The mathematical notation is currently unclear, leading to confusion. There are no prior definitions for the variables and terms used in the equation, resulting in potential misunderstandings.

(a) what is the definition of $S_k$ in line 236? A very rough definition appeared in line 242.

(b) What is the meaning of $w^{t+E}$? Does it refer to the weights at epoch $E + t$? Then what does $W_t^{S_k}$ signify? What is the difference between the two kinds of $t$.  What is the meaning of $S_k$ here? The updated weights using $S_k$? This lack of clarity makes the derivation difficult to follow.

2. Assumption 4 may be too strong to generalize to real-world scenarios. It states that 'Assumption 4 ensures that gradients computed on distilled datasets closely approximate those computed on the aggregated dataset, minimizing error accumulation.' However, based on my experience with dataset distillation, the difference between gradients on distilled datasets and the full dataset can be significant, which could lead to substantial error accumulation.

3.  Figure 2 and Figure 3 are too complicated to understand. There is no emphasis on these figures. The figures should be simplified while the captions should be more detailed.

4. The paper only examines MNIST and CIFAR-10 datasets, which limits its relevance and applicability. For a 2024 publication, a broader range of datasets would strengthen its impact and generalizability, particularly with more complex or diverse data reflecting modern challenges.

**Questions:**

1. For the equation in lines 188 to 192, how would it change if $ n_k $ differs for each client? Also, in the second line, $ x_i $ should be written as $ x^i $, consistent with $ x^i_k $ in the first line.

2. Could the authors conduct experiments on at least the CIFAR-100 dataset? Including Tiny-ImageNet would also be beneficial. Notably, recent dataset distillation papers have conducted experiments on larger datasets, such as ImageNet-1K, which would enhance the paper's relevance and demonstrate the method's scalability.

3. How to adjust the client heterogeneity ($\alpha$) in your experiments? Does any other baselines use the same method to adjust client heterogeneity ($\alpha$)? The reported results of baselines in table 1 are produced by you own reproduction or cited from other papers?

---

> ### Author Response · Authors · 2024-11-24
> **Response to Reviewer ESCL**
>
> We thank Reviewer ESCL for devoting time to this review and drawing valuable comments.
>
> > **Weaknesses 1**: *The mathematical notation is currently unclear, leading to confusion. There are no prior definitions for the variables and terms used in the equation, resulting in potential misunderstandings.
> (a): "what is the definition of $S_k$ in line 236? A very rough definition appeared in line 242.''
> (b): " What is the meaning of $\( w^{t+E} \)$? Does it refer to the weights at epoch $\( E + t \)$? Then what does $\( W_t^{S_k} \) $signify? What is the difference between the two kinds of$ \( t \)$? What is the meaning of $\( S_k \)$ here? This lack of clarity makes the derivation difficult to follow.''*
>
> **Response to Weaknesses 1.a:**
>
>  $\mathcal{S}_k$ represents the distilled dataset of client $k$, which can be found in Line 238.
> In the new version of the submission, it is in Line 229.
>
> **Response to weaknesses 1.b:**
>
>  $w^{t+E}$ represents the trajectory of the global model in timestamp t after E local epochs. $\tilde{w}^{t+1} := w^t - \eta \nabla L(w^t)$ represents parameter updating under ideal centralized training. The shift accumulates over local updates such that the trajectory of the global model deviates from the optimal path and consequently leads to sub-optimal performance and unstable convergence. This can be found in Line 194 and Line 198 in our old version of submission. In the new version of submission, it appears in Lines 202-204.
>
> > **Weaknesses 2**: *``Assumption 4 may be too strong to generalize to real-world scenarios. It states that 'Assumption 4 ensures that gradients computed on distilled datasets closely approximate those computed on the aggregated dataset, minimizing error accumulation.' However, based on my experience with dataset distillation, the difference between gradients on distilled datasets and the full dataset can be significant, which could lead to substantial error accumulation''*
>
> **Response to weaknesses 2:**
>
> We would like to clarify that Assumption 4 ensures that the gradients of distilled data from individual clients closely approximate those of the aggregated distilled data, aligning with the objective of gradient matching. Specifically, we assume:
>
> $$
> \mathbb{E}\Big[\big\|\nabla _ w L(w, \mathcal{S} _ k) - \nabla _ w L(w, \mathcal{S})\big\|\Big] \leq \sigma_g,\; \forall w \in \{w^t | w^{t+1}  = w^t-  \eta \nabla L(w, \mathcal{S}), t = 0,...,T_{g-1}\}.
> $$
>
> This assumption only requires aggregated distilled data rather than the full dataset, making it relatively easier to satisfy in practice.
>
> Moreover, our convergence theorem builds upon the assumption in DynaFed [1]:
>
> $$
> \|\nabla L(w, \mathcal{D} _ {\text{syn}}) - \nabla L(w, \mathcal{D})\| \leq \delta \|\nabla L(w, \mathcal{D})\| + \epsilon,
> $$
>
> which requires that, for any model parameter $w$, the gradient difference between the synthetic dataset $\mathcal{D}_{\text{syn}}$ and the real dataset $\mathcal{D}$ is bounded. However, we find this assumption difficult to satisfy for arbitrary $w$, as such tight alignment is rarely achievable.
>
> To address this, we relax the assumption using **Assumptions 3 and 4**. These assume good **local feature matching** and **global gradient matching**, while restricting model parameters to a series of Gaussian ball regions. Compared to DynaFed’s assumption, these relaxed conditions are more practical and achievable in real-world scenarios.
>
> > **Weaknesses 3**: *``Figure 2 and Figure 3 are too complicated to understand. There is no emphasis on these figures. The figures should be simplified while the captions should be more detailed.''*
>
> **Response to Weakness 3:**
>
> New figures have been updated in the paper. Please refer to Figure 2 (Line 108) and Figure 3 (Line 162) in our new version of the submission.
>
> > **Weaknesses 4**: *``The paper only examines MNIST and CIFAR-10 datasets, which limits its relevance and applicability. For a 2024 publication, a broader range of datasets would strengthen its impact and generalizability, particularly with more complex or diverse data reflecting modern challenges"*
>
> > **Q2**: *``Could the authors conduct experiments on at least the CIFAR-100 dataset? Including Tiny-ImageNet would also be beneficial. Notably, recent dataset distillation papers have conducted experiments on larger datasets, such as ImageNet-1K, which would enhance the paper's relevance and demonstrate the method's scalability.''*
>
> **Response to Weakness4 and Q2:**
>
> In addition to MNIST and CIFAR10, we also designed and conducted experiments on two real-world datasets: Digital-Five and Office-Home. These datasets provide diverse, practical challenges that closely resemble real-world applications, allowing us to rigorously evaluate the effectiveness and adaptability of our method. See Table 2 (Line 495).

---

> ### Author Response · Authors · 2024-11-24
> **Response to Reviewer ESCL(continue)**
>
> > **Q1**: *``For the equation in lines 188 to 192, how would it change if $n_k$ differs for each client?''*
>
> **Response to Q1:**
>
> $n_k=|\mathcal{D}_k|$ represents the size of the dataset on client $k$, which can be found in Line 159 in our old version of the submission. In new version, it appears in Line 161. So, $n_k$ does differs from each client.
>
> > **Q3**: *``How to adjust the client heterogeneity ($\alpha$) in your experiments? Does any other baselines use the same method to adjust client heterogeneity ($\alpha$)? The reported results of baselines in table 1 are produced by you own reproduction or cited from other papers?''*
>
> **Response to Q3:**
>
> See Line 423 (old version) and Line 418 (new version) for how to adjust $\alpha$, i.e., ``Client heterogeneity is adjusted using the Dirichlet distribution’s alpha parameter, where smaller values indicate greater heterogeneity``. The reported results of baselines in Table 1 were produced by our own reproduction because we found that FedDM uses real data to generate synthetic images in order to boost their performance. This will cause significant privacy issues in federated learning. Thus, in our work, all of our synthetic images are generated based on random initialization instead of real data, to protect privacy. In the experiments on DigitFive and OfficeHome, the inherent heterogeneity of data within the same category across different domains makes it unnecessary to use $\alpha$ to control their heterogeneity.
>
> Also, Figure 9 (Line 972) shows that our method outperforms FedDM in real data initialization.
>
> [1] Renjie Pi, et al. Dynafed: Tackling client data heterogeneity with global dynamics. In Proceedings of the IEEE/CVF Conference on Computer Vision and Pattern Recognition,pp.12177–12186,2023.

---

> > ### Author Response · Authors · 2024-11-28
> >
> > Thank reviewer ESCL for your valuable feedback on our submission. We have carefully addressed your comments and uploaded a detailed response along with the revised manuscript on OpenReview. We hope you can review our updates, and if they have resolved your concerns, we would greatly appreciate it if you could consider reflecting this in your rating score.
> >
> > If you have any further questions or if our current response does not fully address your concerns, we would be more than happy to discuss them further.
> >
> > Thank you again for your time and thoughtful review.

---

### Official Review · Reviewer_Fvyf · 2024-11-09

**Soundness:** 2
**Presentation:** 3
**Contribution:** 2
**Rating:** 3
**Confidence:** 4

**Summary:**

In this work the authors propose a new method for private synthetic data for federated learning. The method takes advantage of Local Distribution Matching and further propose Global Gradient Matching to mitigate the heterogeneity exacerbated in the LDM stage. The method comes with some formal guarantees and experiments on several real world datasets suggest that the proposed method converges faster compared to prior baselines.

**Strengths:**

- Overall this work is well written.
- The direction of private synthetic data for federated learning is important.
- The combination of LDM and GGM is nice and well explained.
- The method comes with convergence guarantee.

**Weaknesses:**

- The authors should more carefully highlight and discuss the contribution and difference between the proposed method and FedDM. Correct me if I'm wrong: the key new stuff seems to be 1. Sec 4.2 which is the global gradient matching part, 2. introduction of the error ball in Sec 4.1, which is mainly building upon the technique used in FedDM? If so, I suggest the authors to highlight it more clearly in the intro / list of contributions.
- Theorem 2 does not provide the exact convergence rate. $C$ is a term that contains $\mathbb{E}[L(w_{t-1};\mathcal{D})]$. Hence, it's not a constant. Could the authors work out the convergence rate for the proposed algorithm?
- Privacy guarantee is pretty hand-wavy and I have the following questions.
1. What are $\sigma_r$, $\sigma_s$, and $\sigma_d$ respectively?
2. What does the private algorithm look like? From the appendix, the Gaussian mechanism is applied to the gradient and $R_t$ directly. The later looks like an output perturbation but it's unclear where the former DPSGD style update is performed? Is it done during client update or server update?
3. If the gradient privatization is done at client side, I'm wondering why do we need privatization of $R_t$ as isn't that directly followed by post-processing?
- Algorithm description is not clear enough. Is the synthetic data being regenerated for each communication round?
- Figure 4 seems to provide some intuition on communication efficiency vs utility but it's unclear what is the size of the synthetic data (information communicated in the proposed method) vs the size of the actual gradient (information communicated in FedAvg). A more intuitive plot to me is to replace the x-axis to be the amount of information being communicated in size.
- CIFAR 10 accuracy seems abnormally low (compared to Figure 3 in FedDM). Is it due to the use of a less powerful model?
- Figure 4 legend, I guess it's FedDualMatch instead of FedBi-Match?

**Questions:**

See the above weaknesses about questions related to privacy analysis and experiments.

---

> ### Author Response · Authors · 2024-11-24
> **Response to reviewer Fvyf (Weaknesses 1 and 2)**
>
> We thank Reviewer Fvyf for devoting time to this review and drawing valuable comments.
>
> > **Weaknesses 1**: *``The authors should more carefully highlight and discuss the contribution and difference between the proposed method and FedDM. Correct me if I'm wrong: the key new stuff seems to be 1. Sec 4.2 which is the global gradient matching part, 2. introduction of the error ball in Sec 4.1, which is mainly building upon the technique used in FedDM? If so, I suggest the authors to highlight it more clearly in the intro / list of contributions."*
>
> **Response to Weakness 1:**
>
> - *Introduction of global gradient matching component*. This component represents one of the main contributions in our work. As demonstrated by our experiments, removing the gradient matching mechanism leads to a $0.5\sim 1.0$ performance drop in accuracy.
>
> - *Advantages over FedDM - active error ball*. Unlike FedDM, which uses a fixed error ball radius and lacks flexibility in capturing model variations, we propose an *active Gaussian error ball* with an adaptive generalization error radius. Specifically, we define $R_t$ as the largest norm of the difference between model updates, and compute it on server. This radius boosts performance by maintaining balance between overly dispersed sampling points (which lead to insufficient feature distribution matching among clients) and radiuses that are too small to cover the model space.
>
> - *Advantages over FedDM - layer-wise alignment*. FedDM aligns the synthetic and real data with only the last feature layer, failing to capture intermediate feature representations that are crucial for generalization. In order to fully exploit the feature distribution of data on each client, in addition to the dynamic computation of the generalization error ball’s radius, we put forward the backward layer-wise feature alignment, and incorporate it into the training process. We align each layer’s feature distributions sequentially, starting from the last layer, i.e., the logit layer. This holistic approach captures the intermediate representations crucial for generating synthetic data that better match the real data, resulting in improved performance and generalization.
>
>
>
> > **Weaknesses 2**: *Theorem 2 does not provide the exact convergence rate. $C$ is a term that contains $ E\left [ L(w_{t-1}; \mathcal{D}) \right ]$. Hence, it's not a constant. Could the authors work out the convergence rate for the proposed algorithm?*
>
> **Response to Weakness 2:**
>
> Thank you very much for pointing it out. We followed your suggestion and have updated the convergence result in our paper. Building upon the existing proof for a single round on the server, we extended the convergence analysis to multiple communication rounds and established a convergence theorem, which can be seen in page 8 in new version paper. This theorem shows that the sub-optimality gap of the final model is bounded by a constant primarily determined by the local feature matching bound in Assumption 3. These theoretical findings align with and support the efficiency theorem, illustrating how the effectiveness and convergence of our FedDualMatch method are closely tied to the performance of local data distillation. While our method excels in scenarios with significant heterogeneity, its performance in more homogeneous settings reflects the dependence on advancements in dataset distillation methods executed on clients.

---

> ### Author Response · Authors · 2024-11-24
> **Response to Reviewer Fvyf (Weaknesses 3, 4, and 5)**
>
> > **Weaknesses 3**: *Privacy guarantee is pretty hand-wavy and I have the following questions.*
>
> >*1: ``What are $\sigma_r, \sigma_s$ and $ \sigma_d$ respectively"?*
>
> >*2: ``What does the private algorithm look like? From the appendix, the Gaussian mechanism is applied to the gradient and $R_t$ directly. The latter looks like an output perturbation but it's unclear where the former DPSGD style update is performed? Is it done during client update or server update?"*
>
> >*3: ``If the gradient privatization is done at client side, I'm wondering why do we need privatization of $R_t$ as isn't that directly followed by post-processing"*
>
> **Response to Weaknesses 3.1:**
>
> $\sigma_r, \sigma_s$ are standard deviations of Gaussian noises added to $R_{t}$ and $\mathcal{S}_k$, respectively, which can be found in Line 412.
> The expected feature distance between them is bounded by a small constant  $\sigma _ d > 0 $ for any sampled $w$, which can be found in Line 352.
>
> In the revised manuscript, you can find our algorithm converges with the sub-optimality gap of the final model bounded by a constant determined by the distributional distance $\sigma_d$ in Line 392.
> $\sigma$ is one constant factor on the Gaussian noise variance. This sentence can be found in Algorithm 1, in Line 274, which represents $\sigma_s$.
> For $\sigma_r$, please refer to Response to Weaknesses 3.3.
>
> **Response to Weaknesses 3.2:**
>
> The Gaussian noise is added to the distilled data updating process in Algorithm 1 as follows (see also Lines 292-297). Given clipped gradients:
> $$
>  \nabla _ {\mathcal{S} _ k} L _ {DM}^{k,p}(\mathcal{S} _ k; \mathcal{D} _ k) \leftarrow \frac{\nabla _ {\mathcal{S} _ k} L _ {DM}^{k,p}(\mathcal{S} _ k; \mathcal{D} _ k)}{\max\left(1, \frac{\|\nabla _ {\mathcal{S}_k} L _ {DM}^{k,p}(\mathcal{S} _ k; \mathcal{D} _ k)\| _ 2}{C}\right)} ,
> $$
>
> we add Gaussian noise to them:
>
> $$
>     \nabla _ {\mathcal{S} _ k} L _ {DM}^{k,p}(\mathcal{S} _ k; \mathcal{D} _ k) \leftarrow \nabla _ {\mathcal{S} _ k} L _ {DM}^{k,p}(\mathcal{S} _ k; \mathcal{D} _ k) + \frac{1}{|\mathcal{S}^k|} \mathcal{N}(0, \sigma^2 C^2 \mathbf{I}),
> $$
>
> where $|\mathcal{S}^k|$ is the sample batch size, $\sigma$ is one constant factor on the Gaussian noise variance, and $C$ is the gradient norm bound. This step happens in the client update after each iteration computes the MMD.
>
> As for the Gaussian mechanism is applied to $R_t$, please refer to Response to Weaknesses 3.3.
>
> **Response to Weaknesses 3.3:**
>
> Thank you very much for bringing this to our attention. On second thoughts about the privatization of $R_t$, i.e., the Gaussian error ball radius, in fact, $R_t$ is privacy-insensitive since it has nothing to do with the client data. Thus, in the revised manuscript, we remove the privatization of $R_t$, and only consider privatization with synthetic data which may be sensitive.
>
>
> > **Weaknesses 4**: *``Algorithm description is not clear enough. Is the synthetic data being regenerated for each communication round?''*
>
> **Response to Weakness 4:**
>
>  See the Line of Algorithm in Page 6, i.e., ``Initialize $\mathcal{S}_k$ with random Gaussian noise``. Synthetic data is regenerated from random initialization in each round. The server then uses it to train its own global model. This is consistent with baselines.
>
> > **Weaknesses 5**: *``Figure 4 seems to provide some intuition on communication efficiency vs utility but it's unclear what is the size of the synthetic data (information communicated in the proposed method) vs the size of the actual gradient (information communicated in FedAvg). A more intuitive plot to me is to replace the x-axis to be the amount of information being communicated in size."*
>
> **Response to Weakness 5:**
>
> Figure 6 (Line 918 in revised submission) shows how the size of the synthetic data and that of the model weight change with rounds. We can see that the communication efficiency improves while transferring the synthetic data (images) which is 2.5 times smaller than model weights.

---

> ### Author Response · Authors · 2024-11-24
> **Response to Reviewer Fvyf (Weaknesses 6 and 7)**
>
> > **Weaknesses 6**: *``CIFAR 10 accuracy seems abnormally low (compared to Figure 3 in FedDM). Is it due to the use of a less powerful model?"*
>
> **Response to Weakness 6:**
>
>  In FedDM, real client data is randomly selected as the initialization for synthetic images. In this case, the privacy leakage is very likely to happen with this initialization. For example, a lot of image feature information can be observed in Figure 8 (Line 953 in revised manuscript). Thus, in our experiments, we opted to generate synthetic images from random initialization to prevent potential privacy leakage (see Figure 7 and Line 943). We also conducted additional experiments with real data initialization, where our method still outperforms FedDM (see Figure 9 and Line 972 in revised manuscript).
>
> > **Weaknesses 7**: *``Figure 4 legend, I guess it's FedDualMatch instead of FedBi-Match?''*
>
> **Response to Weakness 7:**
>
> Thank you very much for pointing it out. We corrected it in the revised manuscript, which can be found in Line 453.

---

> > ### Author Response · Authors · 2024-11-28
> >
> > Thank reviewer Fvyf for your valuable feedback on our submission. We have carefully addressed your comments and uploaded a detailed response along with the revised manuscript on OpenReview. We hope you can review our updates, and if they have resolved your concerns, we would greatly appreciate it if you could consider reflecting this in your rating score.
> >
> > If you have any further questions or if our current response does not fully address your concerns, we would be more than happy to discuss them further.
> >
> > Thank you again for your time and thoughtful review.

---

### Author Response · Authors · 2024-11-24
**Summary of paper adjustments**

Dear Senior Area Chairs, Area Chairs, and Reviewers,

We thank all reviewers for their constructive and valuable comments and appreciate the time spent on our manuscript. We have provided responses to the comments to address all reviewer concerns fully. We made minor adjustments to the manuscript in our revised manuscript with changes in blue.

Here's a summary of our adjustments:
- **Differential Privacy Discussion**: Following reviewer Fvyf and bZ42's suggestions, we reduced the emphasis on the differential privacy aspect. Compared to FedDM, our method introduces only one additional step: sharing the Gaussian error ball radius $ R _ t $  between the client and the server. Since $ R _ t $ is privacy-insensitive and independent of client data, FedDualMatch still guarantees $ (\epsilon, \delta) $-differential privacy. To reflect this adjustment, we updated the *abstract (Lines 22-23)*, the *introduction (Lines 78 and 98)*, and the *theoretical section (Lines 407-412)*.

- **Convergence Analysis**:  In response to reviewer Fvyf's comments, we extended the convergence analysis from single-round convergence to global optimization convergence, as detailed in Theorem 2 and its accompanying Remark *(Lines 378-388)*. The global convergence theory demonstrates that our algorithm achieves convergence with the sub-optimality gap bounded by a constant determined by the distributional distance $ \sigma _ d $.  Combined with Theorem 2, this highlights that the effectiveness and convergence of FedDualMatch rely on the accuracy of local client data distillation. We updated the *abstract (Lines 22-23)*, the *introduction (Lines 83-84)*, and the *theoretical analysis (Lines 378-406)* accordingly.

- **Figure Updates**: We replotted Figure 2 *(Lines 108-127)* and Figure 3 *(Lines 162-174)* to better illustrate the method's structure and principles, as per the reviewers' suggestions. Additionally, we added explanatory text *(Lines 123-127)* for clarity. We also corrected the typo in Figure 4 regarding "FedBi-Match," as pointed out by the reviewers *(Line 453)*. Thanks for reviewer mCmd's suggestion.

-  **Algorithm Update**: We incorporated the description and formulas for Gaussian noise-based differential privacy into the algorithm *(Lines 264, 292-298)*. This Gaussian noise is added to the gradients of locally distilled data updates to ensure client data privacy.

- **Preliminary Section**:  To enhance understanding, we revised the presentation in the Preliminary section  *(Lines 178-188)* and added necessary definitions and explanations for $ w _ t^{E} $.

- **Additional Experiments**: Based on all reviewers' suggestions, we added several experiments in the appendix due to space constraints. These include:

  - Comparison of interaction parameter scales *(Lines 972-996)*,
  - Visualization of synthetic data results *(Lines 997-1025)*,
  - Comparisons with real data initialization *(Lines 1026-1045)*,
  - Experiments with different sampling rates *(Lines 1046-1055)*,
  - Computation experiments *(Lines 1056-1073)*,
  - Experiments with more clients *(Lines 1074-1102)*.

We sincerely thank the reviewers for their constructive feedback, which has helped us significantly improve the quality and clarity of the paper. We hope that our revisions and detailed responses effectively address the reviewers' concerns and clarify any misunderstandings. If our responses resolve your concerns about this work, we kindly hope you would consider revisiting and possibly improving the scores. Your support would be greatly appreciated.

---

### Note · Authors · 2025-03-22

I have read and agree with the venue's withdrawal policy on behalf of myself and my co-authors.

---

### Meta-Review · Area_Chair_t5Ar · 2024-12-14

**Metareview:**

This paper proposes FedDualMatch, a dataset distillation method to handle data heterogeneity in federated learning scenarios. FedDualMatch provides for enhanced feature representation and convergence stability. Furthermore, the experiments show some effectiveness on benchmark datasets.

Overall the paper makes a decent contribution. However, I personally feel that the modifications over FedDM are minimal. They are mere tweaks to FedDM, which makes the overall contribution somewhat incremental. The theoretical/convergence results are not surprising, given what has been known. The experiments are conducting on small-scale datasets. Thus, overall, the paper does not pass the bar for acceptance.

**Additional Comments On Reviewer Discussion:**

The authors addressed the reviewer comments by:
(i) Reducing the emphasis on differential privacy;
(ii) Enhancing the convergence analysis;
(iii) Updating the algorithm to incorporate Gaussian mechanism to ensure privacy;
(iv) Conducting additional experiments.

---

### Decision · Program_Chairs · 2025-01-22

Reject